# TOWARDS META-PRUNING VIA OPTIMAL TRANSPORT

**Alexander Theus & Olin Geimer**
{atheus, geimero}@ethz.ch
Department of Computer Science
ETH Zurich, Switzerland

**Friedrich Wicke, Thomas Hofmann[†], Sotiris Anagnostidis[†] & Sidak Pal Singh[†]**
{friedrich.wicke, thomas.hofmann,
sotirios.anagnostidis, sidak.singh}@inf.ethz.ch
Department of Computer Science
ETH Zurich, Switzerland

## ABSTRACT

Structural pruning of neural networks conventionally relies on identifying and discarding less important neurons, a practice often resulting in significant accuracy loss that necessitates subsequent fine-tuning efforts. This paper introduces a novel approach named Intra-Fusion, challenging this prevailing pruning paradigm. Unlike existing methods that focus on designing meaningful neuron importance metrics, Intra-Fusion redefines the overlying pruning procedure. Through utilizing the concepts of model fusion and Optimal Transport, we leverage an agnostically given importance metric to arrive at a more effective sparse model representation. Notably, our approach achieves substantial accuracy recovery without the need for resource-intensive fine-tuning, making it an efficient and promising tool for neural network compression. Additionally, we explore how fusion can be added to the pruning process to significantly decrease the training time while maintaining competitive performance. We benchmark our results for various networks on commonly used datasets such as CIFAR-10, CIFAR-100, and ImageNet. More broadly, we hope that the proposed Intra-Fusion approach invigorates exploration into a fresh alternative to the predominant compression approaches. Our code is available here[1].

## 1 INTRODUCTION

Alongside the massive progress in the past few years, modern over-parameterized neural networks have also brought another thing onto the table. That is, of course, their massive size. Consequently, as part of the community keeps training bigger networks, another community has been working, often in the background, to ensure that these bulky networks can be made compact to actually be deployed (Hassibi et al., 1993). Techniques to reduce the size of these networks and speed-up inference come in various forms, such as pruning — which can itself be unstructured (Han et al., 2015; Singh & Alistarh, 2020), semi-structured (Zhou et al., 2021a), or structured (Wang et al., 2019; Frantar & Alistarh, 2023); quantization (Dettmers et al., 2022; Yao et al., 2022; Xiao et al., 2022); knowledge distillation (Hinton et al., 2015; Gou et al., 2021); low-rank decomposition (Yu et al., 2017); hardware co-design (Zhu et al., 2019), to list a few.

However, despite the apparent conceptual simplicity of these techniques, compressing neural networks, in practice, is not as straightforward as simply doing one or two traditional post-processing steps (Blalock et al., 2020). The process involves a crucial element—fine-tuning or retraining, on the original dataset or a subset—extending over several additional epochs.

While such additional fine-tuning may not seem too much of an issue for some networks, for others like large language models even a single epoch might be excessively expensive. Hence, this makes

---

[†]Advising authors.
[1]Github repository: https://github.com/alexandertheus/Intra-Fusion.

the question of investigating the direction of 'fine-tuning-free' compression methods or even 'data-free' compression methods all the more pertinent.

Besides, since it is almost a given that the training pipeline for taking any interesting network from scratch to deployment will include some form of compression, an overlooked aspect is whether any improvements can be introduced in this joint space of training and pruning in the conventional strategy. For instance, a fine example is the Lottery Ticket Hypothesis (Frankle & Carbin, 2018), which suggests the presence of sparse sub-networks that can be trained from the outset while obviating the need for subsequent pruning. Presently, however, this is more of an existence result, since the sub-networks are obtained retrospectively, i.e., having trained dense networks from scratch. Another prominent example is that of Federated learning (McMahan et al., 2017; Zhang et al., 2021), which has made the community rethink the process of training large models in a distributed manner by carrying local model updates and then aggregating these model parameters directly.

In fact, stemming from the practical interest in federated learning, but also theoretical questions of mode connectivity (Garipov et al., 2018), another novel line of work has lately explored the possibility of fusing (the parameters of) independently trained networks (potentially of different sizes). A notable work in this direction, from which we are heavily inspired, is that of OTFusion (Singh & Jaggi, 2020). More specifically, amongst other things, the authors also demonstrate the idea of fusing a network with a lesser version of itself, say a pruned version, in a bid to help recover the performance drop in the past. However, as their demonstration was merely a proof-of-concept and inherently limited in scope, this exciting idea of self-recovery has remained in a nascent stage.

Our focus in this paper is, therefore, to unite these two lines of work, namely pruning and fusion, in a more cohesive manner. By unifying these two concepts, we sim to expand the horizon of the conventional pruning paradigm — across the trifold axes of:

**(i) Intra-Fusion:** While most research on pruning has focused on devising more meaningful neuron importance metrics, the overlying procedure has remained largely the same: *Keep the most important neurons, discard the rest.* In contrast, Intra-Fusion leverages Optimal Transport to additionally inform the process of model compression with the neurons that otherwise would have been discarded (see Section 3).

**(ii) Data-Free pruning:** Pruning neural networks generally leads to immediate drops in accuracy, hence requiring an extensive fine-tuning step to be usable in a practical setting. Accuracy drops between simpler and more sophisticated neuron importance metrics do not differ substantially. We argue that this is largely an artifact of the underlying pruning procedure and show that, by using *Intra-Fusion, a significant amount of accuracy can be recovered without the need for any datapoints.*

**(iii) Split-Data Training:** Models that are to be pruned after training are most often fine-tuned for many epochs to regain sufficient performance. Via the presented 'PaF' and 'FaP' approaches, we factorize this process through the combination of model pruning and model fusion. By splitting the training dataset into multiple parts, over which models are trained concurrently, we achieve significant training time speedups.

## 2 BACKGROUND

### 2.1 PRUNING

Pruning techniques (LeCun et al., 1989) can broadly be classified into *structured* (Wang et al., 2019; Fang et al., 2023), and *unstructured* pruning (Han et al., 2015; Singh & Alistarh, 2020). Unstructured pruning involves zeroing out individual weights and leaves the network structure unaltered. Our work exclusively deals with *structured pruning*, which aims to remove entire sets of parameters or neurons and thereby directly alters the network structure. The reason being that (a) the structured pruning procedure directly translates into a speedup in the inference time — unlike in unstructured pruning, which requires the aid of specialized hardware accelerators to extract some (typically reduced) levels of speedup; (b) structured pruning eases the storage and memory footprint of the network; while unstructured pruning methods yield no such gains (but, in fact, also necessitate the maintenance of binary masks during the course of training).

**Structured pruning.** A very common and successful approach to prune networks structurally is to capture a neuron's importance by its $\ell_p$-norm, where $p$ is the order of the norm. Despite its simplicity, $\ell_p$-norm pruning can achieve state-of-the-art performance by cleverly incorporating the

dependencies within the network (Fang et al., 2023). However, other works such as (He et al., 2019) have shown that the "smaller-norm-less-important assumption" does not always hold. Instead of removing neurons with a small norm, redundant filters can be found by exploiting relationships between neurons of the same layer. Namely, they consider neurons close to the geometric mean to be redundant as they represent information abundant in the layer. Instead of evaluating importance based on the weight itself, other methods focus on the activations of the neurons. The importance of a neuron is thus measured by evaluating the reconstruction error of the current layer (He et al., 2017) or of the final response layer (Yu et al., 2018).

Evidently, research on structured pruning has largely focused on devising more meaningful importance measures, while the overlying procedure has remained the same, repeating the mantra: *Keep the most important neurons, discard the rest*. This work *challenges* the above mantra by recycling or restoring information from all neurons to create more accurate compressed networks.

## 2.2 OPTIMAL TRANSPORT & MODEL FUSION

**Optimal Transport (OT).** OT (Villani et al., 2009) is a mathematical framework that provides a rigorous and geometrically interpretable way to compare probability distributions. The OT problem aims to find the most economical way to "transport" mass from one distribution defined over a space $\mathcal{X}$ to another supported over the space $\mathcal{Y}$, where the cost is determined by a function $c$. It achieves this by lifting the metric in the ground space (i.e., $\mathcal{X}, \mathcal{Y}$) to obtain a metric in the space of distributions. In the case of discrete probability measures, OT reduces to the well-known transportation problem in linear programming, which has the following form:

$$\text{OT}(\mu, \nu; C) := \min \, \langle T, C \rangle_F \quad \text{s.t.,} \quad T\mathbf{1}_m = \alpha, \, T^\top \mathbf{1}_n = \beta \quad \text{and} \quad T \in \mathbb{R}_+^{(n \times m)} \, .$$

Here, $\mu := \sum_{i=1}^n \alpha_i \cdot \delta(x_i)$ and $\nu := \sum_{j=1}^m b_j \cdot \delta(y_j)$, with $\sum_{i=1}^n \alpha_i = \sum_{j=1}^m \beta_j = 1$ describe two probability measures, where $\delta(\cdot)$ denotes the dirac delta function. Further, $T$ denotes the transport map whose row and columns should sum to the marginals $\alpha$ and $\beta$. Besides, $C$ denotes the ground cost, of moving a unit mass from point $x_i$ to $y_j$, and for instance, in the Euclidean case, it can be $C(x_i, y_j) = \|x_i - y_j\|^2$. Optimal Transport has found applications in many fields, especially in machine learning (Kusner et al., 2015; Frogner et al., 2015; Arjovsky et al., 2017; Zhou et al., 2021b) and we will consider this in regards to fusion.

**Model Fusion.** The key idea behind model fusion is to combine the capabilities of two parent networks into a single-child network. Singh & Jaggi (2020) introduces Optimal Transport (OT) for model fusion, which is the fusion technique we are using throughout this paper. We will refer to this method as *OTFusion*. The main idea behind this work is to combine multiple independently trained neural networks, after accounting for the permutation symmetries that exist within the layers. Finding the permutation symmetries is then framed as an Optimal Transport problem between the neurons of the given networks. An important thing to note is that Singh & Jaggi (2020) primarily use uniform distributions in place of $\alpha$ and $\beta$; however as we will see later, we further exploit this in our proposed method.

Besides *OTFusion*, other works are also inherently based on a similar formulation (Li et al., 2015; Yurochkin et al., 2019; Wang et al., 2020), though not always geared towards the same end. Our focus will nevertheless be on *OTFusion*, as we directly build atop their exploration of pruning and fusion.

## 3 METHODOLOGY

Intra-Fusion, as a "meta-pruning" approach, is an attempt to take a step back from the search for meaningful importance metrics, and reconsider the way these found importance metrics are leveraged to come up with a sparse model representation. Instead of simply discarding the least important neurons (as done in the "conventional pruning" approach, see Algorithm 1), Intra-Fusion leverages a modified version of OTFusion to incorporate the discarded neurons into the "surviving" ones (see Algorithm 2). We refer to our algorithm as a new "meta" approach to pruning, since it challenges the overlying framework that defines how an agnostic importance metric is integrated. It does not compete with any importance metrics. Instead it provides an alternative methodology on how these metrics are used to compress a network.

### 3.1 Meta Pruning: An Overview

This section is dedicated to the inner workings of Intra-Fusion. Since Intra-Fusion is an alternative to the conventional pruning procedure, we show both meta-pruning approaches side by side (see Algorithm 1 and 2) to highlight the differences between the two. First, we present below a high-level overview of the two algorithms, and then the subsequent sections contain more detailed explanations of the individual parts that make up Intra-Fusion, which are also referenced in Algorithm 2.

---

**Data:** IMPORTANCE $\boldsymbol{i}_{1 \times n}$, group $\mathbb{G}$ with group cardinality $n$, and target group cardinality $m$.
**Result:** group $\mathbb{G}_{new}$ with group cardinality $m$

| **Algorithm 1:** Conventional Pruning | **Algorithm 2:** Intra-Fusion | |
|---|---|---|
| $t \leftarrow m^{\text{th}}$ highest scalar in $\boldsymbol{i}$ ; | $\mathbb{Y} \leftarrow \text{GETTARGET}(\mathbb{G})$ ; | 3.2.1 |
| $\mathbb{G}_{new} \leftarrow \emptyset$ ; | $\mathbb{X} \leftarrow \mathbb{G}$ ; | 3.2.1 |
| **for** $layer \in \mathbb{G}$ **do** | $\boldsymbol{C}_{n \times m} \leftarrow \text{COMPUTECOST}(\mathbb{X}, \mathbb{Y})$ ; | 3.2.2 |
|    $layer_{new} \leftarrow layer$ without neurons $j$ | $\boldsymbol{\nu}_{1 \times m} \leftarrow \text{GETTARGETDISTR}(m, \boldsymbol{i})$ ; | 3.2.3 |
|         where $\boldsymbol{i}[j] < t$ ; | $\boldsymbol{\mu}_{1 \times n} \leftarrow \text{GETSOURCEDISTR}(n, \boldsymbol{i})$ ; | 3.2.3 |
|    $\mathbb{G}_{new} = \mathbb{G}_{new} \cup \{layer_{new}\}$ ; | $\boldsymbol{T}_{n \times m} \leftarrow \text{OT}(\boldsymbol{\mu}, \boldsymbol{\nu}, \boldsymbol{C})$ ; | 3.2.4 |
| **end** | $\boldsymbol{T}_{m \times n} \leftarrow \text{diag}(\frac{1}{\boldsymbol{\mu}}) \times \boldsymbol{T}^{\top}$ ; | 3.2.4 |
| | $\mathbb{G}_{\text{new}} \leftarrow \emptyset$ ; | |
| | **for** $layer \in \mathbb{G}$ **do** | |
| |    $\mathbb{G}_{new} = \mathbb{G}_{new} \cup \{\boldsymbol{T} \times layer\}$ ; | 3.2.4 |
| | **end** | |

---

**Structured Pruning: Group-by-Group.** As described in Section 2.1, the increasing complexity of neural networks pose a challenge for structured pruning. Removing neurons from layers individually can lead to broken networks, as neurons in a layer might not only be dependent on the previous layer, but also on those that lie even further back.

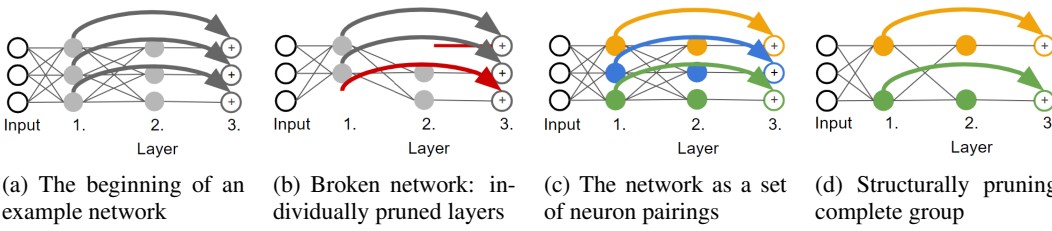

(a) The beginning of an example network

(b) Broken network: individually pruned layers

(c) The network as a set of neuron pairings

(d) Structurally pruning complete group

Figure 1: Structural pruning by considering groups

In Figure 1, we show an example of this: Assume we are given the beginning of a network in Figure 1a, where layer three represents the arrival of a residual connection. When iterating through the network layer-by-layer, while pruning, the network can easily be broken (see Figure 1b). As multiple layers can be arranged in such a dependency, pruning of multiple layers has to be done jointly (in our example layers one to three). We call a set of layers that have to be pruned in unison a group $\mathbb{G}$. In a group, not all neurons are dependent on one another. We can identify "pairings of neurons" that have to be handled jointly (see the different colored pairings in Figure 1c). The number of neuron pairings in a group, we term "group cardinality" (in our e.g., this would be three).

Structurally pruning the network in Figure 1a whilst considering the dependencies could result in the network shown in Figure 1d.

**Meta-Pruning Comparison.** The starting point for conventional pruning and for our Intra-Fusion methodology is the same. We are given a group $\mathbb{G}$ with initial group cardinality $n$, that we wish to prune to a target group cardinality $m \leq n$. Moreover, we are given an importance vector $\boldsymbol{i}_{1 \times n}$ that assigns an agnostic importance score (e.g. $\ell_1$-norm) to each independent pairing in the group. In conventional pruning (see Algorithm 1), we keep the $m$ most important neuron pairings according to $\boldsymbol{i}$, and remove the rest to arrive at our new group $\mathbb{G}_{new}$ with group cardinality $m$.

Most research on structured pruning has focused on devising more meaningful importance measures, i.e. $\boldsymbol{i}$, whereas the overlying procedure detailed in Algorithm 1 has remained practically unaltered. Inspired by *OTFusion*, we attempt to develop an alternative to the way pruning is conventionally done. Instead of simply discarding the less important pairings in a group (in Figure 1c this would be the blue pairing), we leverage the computed importance metrics to inform the process of fusing these pairings to end up at a lower group cardinality.

To end up at a lower group cardinality, we fuse the exisiting $n$-many pairings to end up at $m$-many pairings. The matching of the pairings to be fused is found via Optimal Transport (OT), which requires the careful setting of multiple non-trivial hyper-parameter choices. In particular, the discrete "source distribution" of our OT problem is representative of the original network's group $\mathbb{G}$ (group cardinality $n$), i.e., it is supported on the space of neuron pairings.

To complete the OT problem formulation we additionally need to determine a target of group cardinality $m$, and a neural similarity measure to quantify the transportation cost. Lastly, we need to ascertain the probabilistic measures encapsulating the mass distribution of both the source and target entities. Solving the just formulated OT problem gives a transportation map $\boldsymbol{T}$.

### 3.2 COMPONENTS OF INTRA-FUSION

3.2.1 TARGET AND SOURCE SELECTION. Our experiments reveal that a simple, but fruitful option is to use $\mathbb{G}_{new}$ derived by Algorithm 1, i.e. the neurons with the highest importance according to some agnostically defined metric. Another promising approach is to cluster the neurons with K-means (Lloyd, 1982), or Gaussian Mixture Models, with $m$ clusters, and use the respective cluster centroids as the target.

3.2.2 TRANSPORTATION COST. Once we have determined the source and the target, the next step is to quantify the cost of moving a unit mass from a source neuron to a target neuron. For this we will take a metric that measures how similar or dissimilar neurons are. In case a group contains multiple layers (as shown in Figure 1), we have to find a joint similarity measure for pairings of neurons.

Given two vectors $\boldsymbol{a}$ and $\boldsymbol{b}$, each representative of the weights of a different neural pairing, we determine similarity by their normalized $\ell_1$-distance. The weights for neural pairings can be derived via concatenating the weights of the respective neurons and bias terms.

However, architectural components such as Batch Normalization (BN) (Ioffe & Szegedy, 2015) evidently present themselves as a challenge due to their unique connection with their prior layer. We overcome this challenge by taking advantage of the properties of batch normalization, and simply merge the batchnorm layer into the prior layer whose outputs it acts upon (Jacob et al., 2018).

$$\boldsymbol{w}_{new} = \frac{\boldsymbol{w} \times \gamma}{\sqrt{\sigma}}, \quad b_{new} = \frac{(b - \mu) \times \gamma}{\sqrt{\sigma}} + \beta \,. \tag{1}$$

In this context, $\boldsymbol{w}$ denotes the weight vector corresponding to the preceding layer. Furthermore, we denote $b$ as its associated bias term. The symbols $\mu$ and $\sigma$ represent the running mean and variance, respectively, of the batch normalization layer, while $\gamma$ and $\beta$ symbolize the learnable parameters.

3.2.3 PROBABILITY DISTRIBUTION. We propose two different ways of quantifying probability mass: uniform or importance-informed. A uniform distribution prescribes equal mass to each neuron pairing. In the importance-informed option, the mass is relative to the importance of the neuron pairing. In order to transform the importance of a neuron pairing to a probability, one can either divide the importance by the sum of all importances, or use softmax. In Appendix C.3 we show that there are no significant differences in accuracy between the choice of source and target distribution.

3.2.4 DERIVING FUSED NEURONS. Given the cost matrix $\boldsymbol{C}$, and our probability distributions $\boldsymbol{\mu}$, and $\boldsymbol{\nu}$, we can finally derive the optimal transport map $\boldsymbol{T}$. Since the columns of this transport map act as the coefficients for the weighted aggregation of corresponding neuron pairings, it is imperative to ensure that these coefficients collectively sum to unity.

Moreover, as a final preparatory step before amalgamating matching neuron pairings, it is necessary to conjoin the batch normalization layer with the layer upon which it operates. This process is elucidated in Equation 1. Subsequently, we establish the values of $\mu$ and $\beta$ as zero, and $\sigma$ and $\gamma$ as one, thereby preserving the unaltered state of the batch normalization layer's activation.

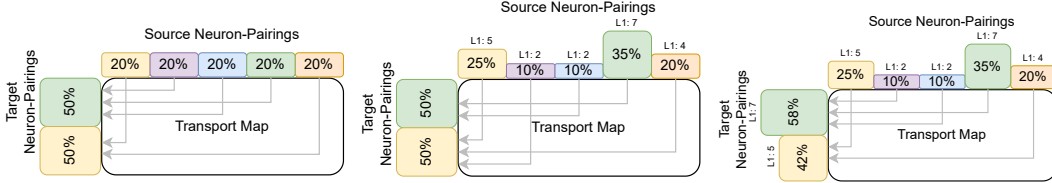

(a) Uniform target and source distribution.   (b) Uniform target and importance-informed source distribution.   (c) Importance-informed target and source distribution.

Figure 2: Options for target/source probability mass distribution with $\ell_1$-norm as importance.

Consequently, we proceed to traverse each layer within the group denoted as $\mathbb{G}$, and rather than removing neurons of diminished significance, we opt to generate fused neurons through a process of matrix multiplication with the transport map $T$. This methodology culminates in pruned layers that emerge as a product of a nuanced aggregation, where the shared features of these neurons are subject to a weighted summation.

## 4 EMPIRICAL RESULTS

Here, we seek to illustrate the accuracy gains Intra-Fusion can achieve. Most pruning literature ignores pre-finetuning accuracy, largely due to the significant accuracy drops imposed. In Section 4.1, we show that this drop is mainly an artefact of the overlying pruning methodology. Namely, by utilizing importance metrics in a more involved way as done in Intra-Fusion, a significant amount of accuracy can be maintained. For the sake of completeness, we also show how Intra-Fusion stands up in face of fine-tuning in Appendix E.2.

**Terminology.** Since we cut whole neurons and not just individual edges we will distinguish between "neuron sparsity" and "weight sparsity". We refer to "neuron sparsity" as the proportion of neuron pairings that are cut out of a group. Accordingly, we will refer to the proportion of edges removed from the neural network as "weight sparsity". Where unspecified, we are referring to neuron sparsity when talking about "sparsity". See Appendix E.1 for a comprehensive comparison.

Lastly, in this section we use the term "Group". This refers to $\mathbb{G}$ as described in Section 3. Moreover, the group indices are ordered such that small indices are closer to the output of the model, e.g. Group 4 is closer to the end of the model than Group 5.

### 4.1 DATA-FREE: PRUNING WITHOUT FINE-TUNING

In order to compare the data-free performance of the conventional meta-pruning paradigm (see Algorithm 1), and Intra-Fusion (see Algorithm 2), we compare the test accuracy of a VGG11-BN, ResNet18, ResNet50, on CIFAR-10, CIFAR-100, and ImageNet. Furthermore, the pruning is done on the basis of a group, for importance metrics $\ell_1$, and more sophisticated ones such as Taylor (Molchanov et al., 2019), LAMP (Lee et al., 2021) and CHIP (Sui et al., 2021). Given the nature of Taylor importance, additional information about parameter gradients is needed when deploying it as a metric. Due to the extensive set of experiments, we are forced to show only a selection in this section and refer to Appendix E.3 for a more comprehensive list of results.

A snapshot of the results can be seen in Figure 3 and 4. While diverse importance metrics do not seem to affect pre-finetuning accuracy meaningfully, Intra-Fusion can leverage an importance metric to substantially increase accuracy (by up to +60% in certain cases) without any additional use of data. To further highlight that the improvements of Intra-Fusion are agnostic to the choice of importance metric, we include results assigning random scores drawn from a uniform distribution as an importance metric (see Appendix C.4).

Across different network architectures, there appear to be, in general, two different kinds of groups: volatile and resilient. Volatile groups exhibit strong accuracy losses as the sparsity increases (e.g., see "Group 6" in Figure 4), whereas resilient groups only experience small ones (e.g., see "Group 16" in Figure 3). This pattern seems to be agnostic with respect to the importance metric. Nevertheless, Intra-Fusion manages to increase accuracy substantially for both types.

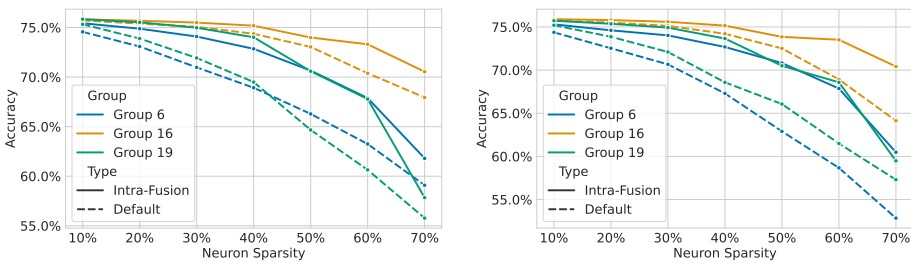

Figure 3: Data-Free Pruning: ResNet50 on ImageNet. $\ell_1$ (left), Taylor (right).

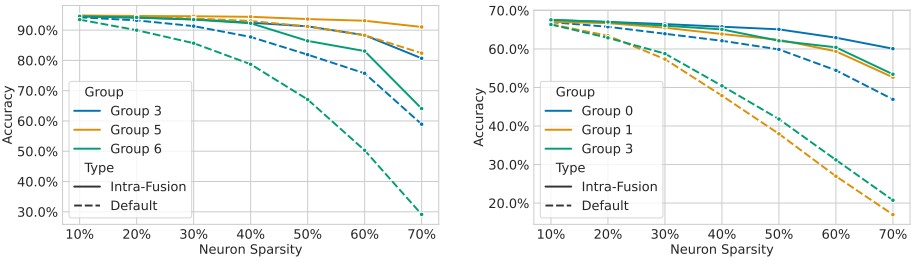

Figure 4: Data-Free Pruning: ResNet18 on CIFAR-10 and VGG11-BN on CIFAR-100, $\ell_1$.

Take for instance Group 6 from Figure 4 (left), a volatile group. For a neuron sparsity of 40%, the conventionally pruned model has dropped to an accuracy of only 78.7%, whereas the Intra-Fused model is still at a competitive 92.2%. However, even for resilient groups, where the margin of improvement is low, we make improvements (see Group 16 in Fig. 3). These results represent a general trend (see Appendix E.3 for all results). *Overall, this shows the benefit of our proposed approach, which can alleviate the performance drop without relying on fine-tuning, or for that matter on any datapoints at all.*

## 5 UNDERSTANDING INTRA-FUSION

So far, we have elucidated and juxtaposed Intra-Fusion with the conventional pruning methodology, demonstrating that Intra-Fusion possesses the distinctive capability to enhance accuracy significantly without relying on any data. In this section, we would like to obtain a better understanding of the inner workings of Intra-Fusion. For a more comprehensive analysis, please see Appendix C.

### 5.1 OUTPUT PRESERVATION

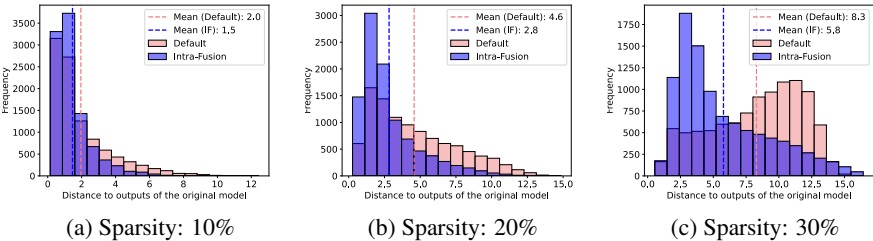

(a) Sparsity: 10%    (b) Sparsity: 20%    (c) Sparsity: 30%

Figure 5: Output preservation comparison of the original model. Model: ResNet18. Dataset: CIFAR-10. Group: 0. Importance metric: $\ell_1$.

A very intuitive explanation for the superior performance of Intra-Fusion is its ability to better preserve the output of the original non-pruned model. Hence, we quantify output divergence by the $\ell_2$-distance to the output of the original model for various groups in the data-free scenario at different sparsities. As can be seen in Figure 5 (and in more detail in Appendix C.1), Intra-Fusion is indeed able to preserve the output better, with the margin growing as the sparsity increases. Thus, it

seems that merging akin neurons as we do with Intra-Fusion leads to better output preservation, and subsequent superior performance.

## 5.2 NEURAL LOSS LANDSCAPE

To gather further insights as to how Intra-Fusion differs from the conventional pruning methodology when navigating the weight space, we show an example of a pruned network and the corresponding accuracy landscape that surrounds the models. We vectorize all the parameters before carrying out their linear interpolation for this figure and following the procedure in (Garipov et al., 2018). We specify three models in that space (in our case: original model, default pruned model, and the Intra-Fusion model), one of which serves as the origin, and thus a 2D slice is built. In this slice, we sample a grid of possible networks and evaluate their performance on the test set. It is important to note that not all models in the given 2D slice of the parameter space are pruned (e.g. original model).

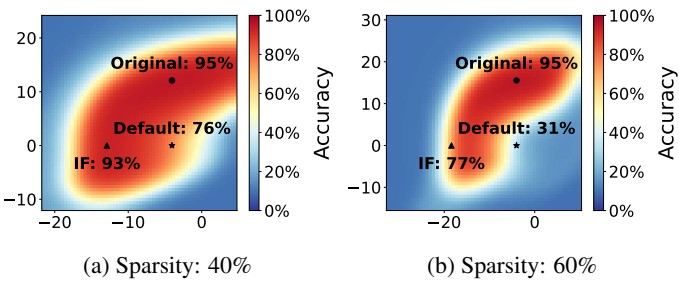

(a) Sparsity: 40%       (b) Sparsity: 60%

Figure 6: Resnet18. CIFAR-10, when group 6 is pruned. IF: Intra-Fusion.

In Figure 6 we depict how the accuracy varies for different models in the identified 2D slice of the parameter space. Evidently, the Intra-Fusion model ends up at a more convenient part of the accuracy landscape when compared to the regularly pruned model ("Default") resulting in a superior performance. To further explore how the accuracy landscape develops through different sparsities and when the whole model is pruned, we include additional figures in Appendix C.2. It appears that using the less important neurons to inform the process of model pruning is more effective at identifying a favorable spot in the parameter space than simply discarding them.

## 6 APPLICATION BEYOND PRUNING: FACTORIZING MODEL TRAINING

In an attempt to find further valuable integrations of pruning and fusion, we also explore how fusion can be used to "factorize" and possibly speed up the training process of models that are supposed to be pruned after training. In an increasingly digitalized world the amount of available data points is consistently increasing. We are looking for an approach that manages to leverage the fact that also on subsets of the whole dataset, a competitive performance can be achieved. This factorization provides another angle on enhancing the performance and training time of models. It could for example be especially interesting as an alternative or enhancement to data parallelism during distributed model training.

The model that is created with the standard pruning approach (trained on the whole dataset, then pruned and finally fine-tuned) we call the "whole-data model". Like in a real-world setting, the result of the pruning is fine-tuned since this most of the time recovers a lot of performance.

## 6.1 THE SPLIT-DATA APPROACH

In the approach we want to propose, we split the model training into smaller phases by utilizing pruning and fusion. For this we first split the data set into two subsets $a$ and $b$ on which we then train two individual models $model_a$, $model_b$ in parallel. This leads to a theoretical 2x speedup in the model training time since the convergence on half of the dataset does not take more epochs than on the whole dataset (see Figure 20). These two models are then fused using OT and fine-tuned on the whole data set. To reach the target sparsity we prune the resulting network and fine-tune again. We call this approach FaP (Fuse and Prune, see Figure 7). For completeness we also include the performance of individually pruning and fine-tuning the networks before fusing — this we call PaF

(Prune and Fuse, see Figure 21). For the pruning part of doing FaP and PaF, we use the introduced Intra-Fusion.

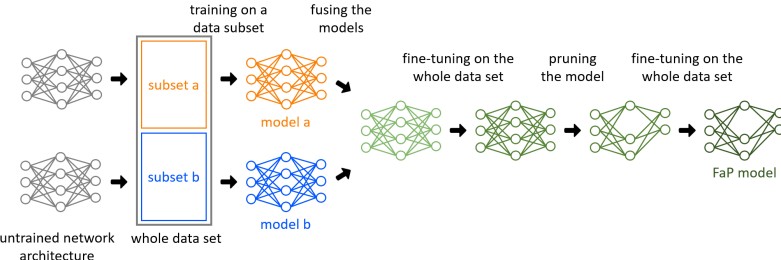

Figure 7: An illustration of the Fuse and Prune (FaP) approach.

To give a concrete insight into the relative runtimes of the different approaches, we give a side-by-side comparison of their timelines in Figure 22. Relative to the model training time of a VGG11-BN (CIFAR-10), we achieve a speedup of 1.81. The overall speedups of the split-data approaches are 1.42 (PaF) and 1.31 (FaP). Importantly, in applications, the time for training $T_1$ of a model on large datasets will typically be much greater than the time for fine-tuning $T_2$ ($T_1 \gg T_2$), and speedups are expected to be much more significant.

## 6.2 SPLIT-DATA PERFORMANCE

Deploying the Split-Data approach when training Resnet18 on CIFAR-10, we were able to recover and even slightly improve over the "whole-data model" performance, while providing a significant speedup in the training time of the involved models (see Appendix D.2). For VGG11-BN we achieve similar results at the cost of higher resource requirements ("k-Fold" setting, see Appendix D.4).

We delve deeper into the details and performance of this "Split-Data" concept in Appendix D. Specifically see "Performance Comparison: After Convergence" (Appendix D.2) and "Performance Comparison: Varying Fine-Tuning" (Appendix D.3). An extension to combining more than the presented two datasets can be found in "k-Fold Split-Data" (Appendix D.4).

## 6.3 SPLIT-DATA AS ALTERNATIVE TO DATA PARALLELISM

Splitting the dataset into different parts that models are trained on individually is not a new idea. Specifically, during distributed model training in cloud infrastructures, this is a common approach called **Data-Parallelism**. However, the gradients are exchanged among the models after the individual backpropagation steps so all models are effectively updated with gradients computed from the whole dataset. This leads to high network utilization, sensitivity to network latency and wait times between the training steps.

In our **Split-Data** approaches (like PaF and FaP), we completely separate the model training. Each model has its designated part of the training data it is trained on. Only after the models have individually converged are the edge weights communicated and fused. This yields far less communication overhead and is not sensitive to network latency.

## 7 CONCLUSION

In sum, we perform a detailed investigation of unifying and bridging the paradigms of pruning and fusion through our conceptions of Pruning-and-Fusion as well as Fusion-and-Pruning. Specifically, we showed how our proposed technique of Intra-Fusion provides a consistent gain — with and without fine-tuning, the latter being also privacy-preserving and highly cost-efficient. We also investigated how fusion can be used to factorize the training process, given that it is subsequently accompanied by pruning, to result in non-trivial speedup in training times. The sparsity obtained via our algorithm is also amenable to actual speedup in inference times, without needing special accelerators. Overall, our work shows the compatibility of bringing together pruning and fusion in the form of meta-pruning, and the potential inherent therein. All in all, this raises the question of rethinking the pre-dominant paradigm and perhaps redefining our approach to obtaining compact models via fusion.

ACKNOWLEDGEMENTS

Sidak Pal Singh would like to acknowledge the financial support from Max Planck ETH Center for Learning Systems.

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

# Appendix

## Table of Contents

# A    EXPERIMENTAL HYPERPARAMETERS

## A.1    TRAINING

During the training of the used VGG11-BN and Resnet18 networks, we deploy the training hyperparameters in Table 1. For the fine-tuning of models after pruning we use the hyperparameters in Table 2.

Table 1: Hyperparameters during model training.

| Loss function | Cross Entropy |
|---|---|
| Optimizer | SGD with momentum = 0.9 |
| Learning Rate Schedule | $0.05 \times 0.5^{\lfloor epoch/30 \rfloor}$ |
| Training Epochs | 300 |
| Batch Size | 128 |

Table 2: Hyperparameters during fine-tuning.

| Loss function | Cross Entropy |
|---|---|
| Optimizer | SGD with momentum = 0.9 |
| Learning Rate Schedule | $0.01 \times 0.5^{\lfloor epoch/30 \rfloor}$ |
| Batch Size | 128 |

## A.2    INTRA-FUSION SETTINGS

For our data-free and data-driven results, we use a homogeneous distribution for both the target and source distribution. Moreover, we use the most important neuron pairings as the target.

# B    IMPLEMENTATION

As mentioned in Section 3, structural pruning is inherently complex due to the inderdependencies existing within state-of-the-art neural networks. Researchers and engineers have relied on manually-designed and model-specific schemes to handle these. Evidently, this is intractable and not scalable, particularly for more complex networks.

Recently, (Fang et al., 2023) introduced a fully-automatic and general way to structurally prune neural networks, by extracting a dependency graph from the computational graph derived by back-propagation. Thus, in order for Intra-Fusion to be generally applicable across a wide range of models in an automized fashion, we have extended their library to work with Intra-Fusion. This way, Intra-Fusion can be applied to a wide and diverse set of models without the user having to adapt the code in any way.

# C    UNDERSTANDING INTRA-FUSION

In the following sections, we want to further expand on Section 5.1 by providing more extensive results and background information.

## C.1    OUTPUT PRESERVATION

The following figures show how well Intra-Fusion is able to preserve the output of the non-pruned model for VGG11 and ResNet18 on CIFAR-10 and CIFAR-100. As before, Intra-Fusion seems to be able to better preserve the output at low to high sparsities.

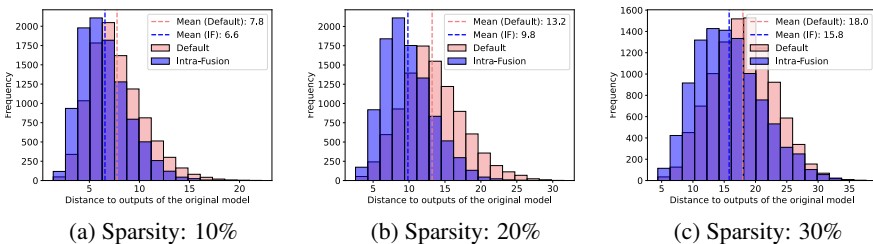

(a) Sparsity: 10%  (b) Sparsity: 20%  (c) Sparsity: 30%

Figure 8: Output preservation comparison of the original model. Model: VGG11. Dataset: CIFAR-10. Group: 0. Importance metric: $\ell_1$.

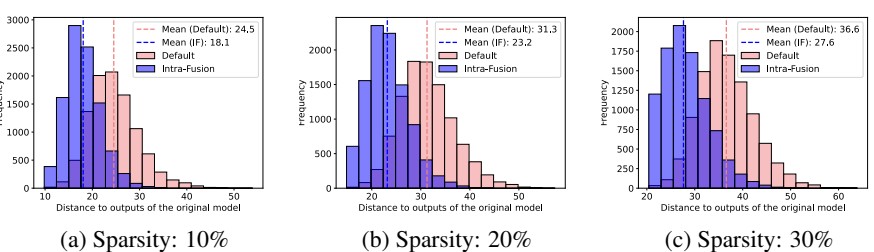

(a) Sparsity: 10%  (b) Sparsity: 20%  (c) Sparsity: 30%

Figure 9: Output preservation comparison of the original model. Model: ResNet18. Dataset: CIFAR-100. Group: 0. Importance metric: $\ell_1$.

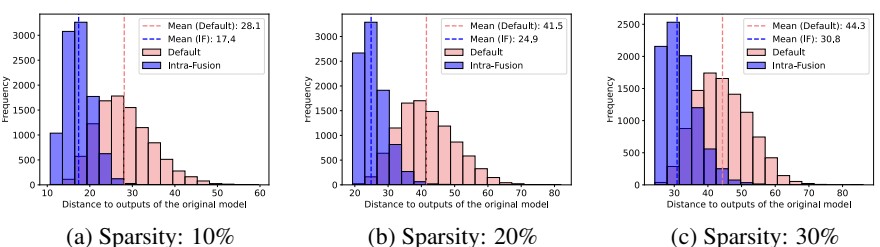

(a) Sparsity: 10%  (b) Sparsity: 20%  (c) Sparsity: 30%

Figure 10: Output preservation comparison of the original model. Model: VGG11. Dataset: CIFAR-100. Group: 0. Importance metric: $\ell_1$.

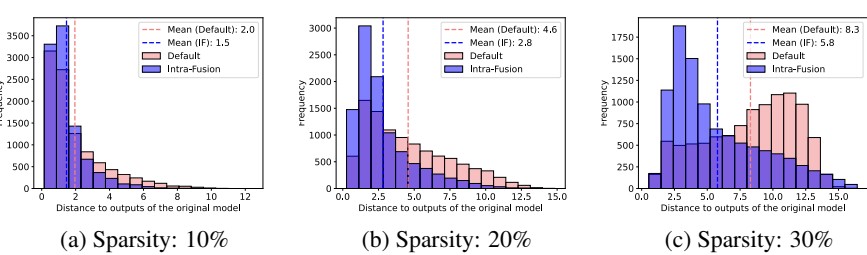

(a) Sparsity: 10%  (b) Sparsity: 20%  (c) Sparsity: 30%

Figure 11: Output preservation comparison of the original model. Model: ResNet18. Dataset: CIFAR-10. Group: 1. Importance metric: $\ell_1$.

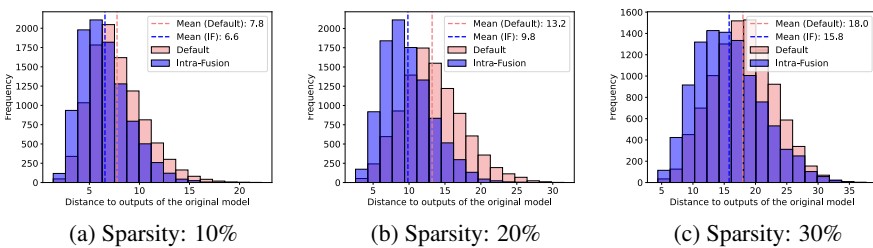

(a) Sparsity: 10%  (b) Sparsity: 20%  (c) Sparsity: 30%

Figure 12: Output preservation comparison of the original model. Model: VGG11. Dataset: CIFAR-10. Group: 1. Importance metric: $\ell_1$.

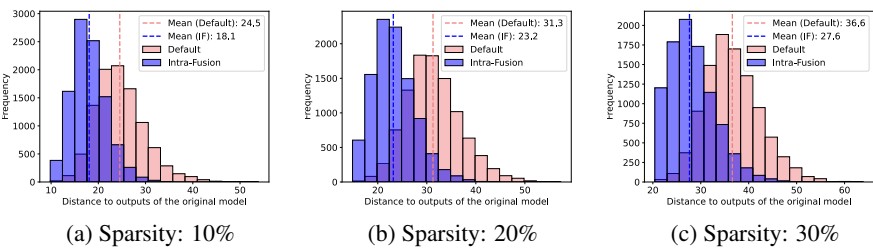

(a) Sparsity: 10%  (b) Sparsity: 20%  (c) Sparsity: 30%

Figure 13: Output preservation comparison of the original model. Model: ResNet18. Dataset: CIFAR-100. Group: 1. Importance metric: $\ell_1$.

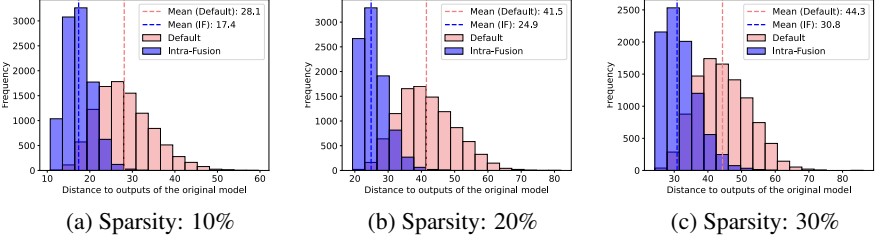

(a) Sparsity: 10%  (b) Sparsity: 20%  (c) Sparsity: 30%

Figure 14: Output preservation comparison of the original model. Model: VGG11. Dataset: CIFAR-100. Group: 1. Importance metric: $\ell_1$.

## C.2 Neural Landscape

We expand the experiment in Section 5.1 by showing results for more sparsities and model-dataset combinations.

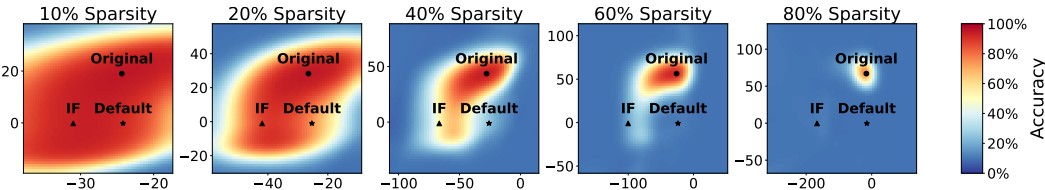

Figure 15: Slice of the model weight space: Resnet18, CIFAR-10, neuron sparsity: 10%-80%, criterion: $\ell_1$, no fine-tuning. IF: Intra-Fusion.

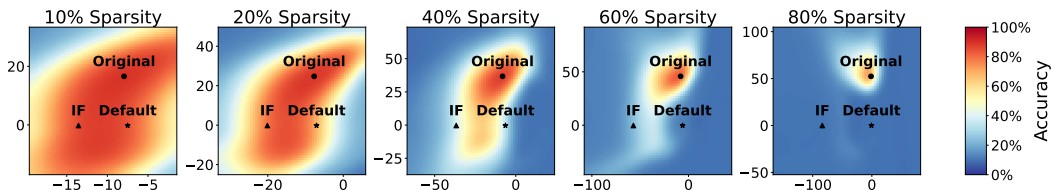

Figure 16: Slice of the model weight space: VGG11-BN, CIFAR-10, neuron sparsity: 10%-80%, criterion: $\ell_1$, no fine-tuning. IF: Intra-Fusion.

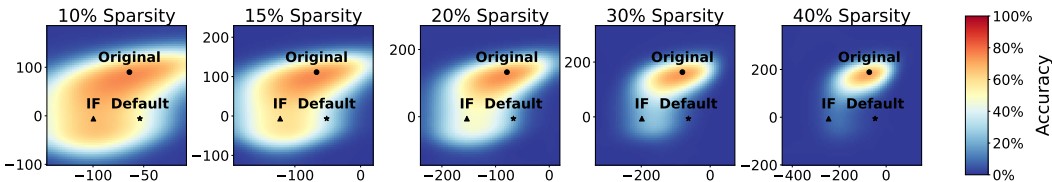

Figure 17: Slice of the model weight space: Resnet18, CIFAR-100, neuron sparsity: 10%- 40%, criterion: $\ell_1$, no fine-tuning. IF: Intra-Fusion.

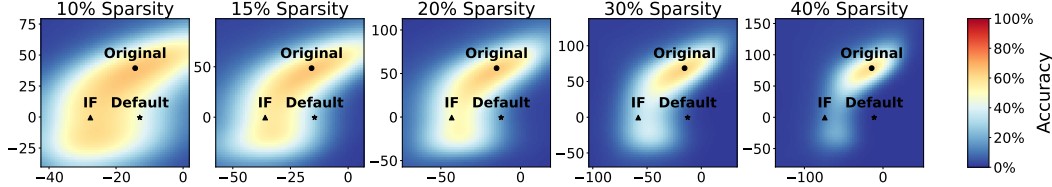

Figure 18: Slice of the model weight space: VGG11-BN, CIFAR-100, neuron sparsity: 10%-40%, criterion: $\ell_1$, no fine-tuning. IF: Intra-Fusion.

## C.3 Ablation Study on Varying Target and Source Distribution

In this section, we intend to shed a light on the potential differences between choosing uniform or an importance-informed distribution for the target and source distribution, as explained in Section 3.2.

In our analysis, we trained six ResNet18 models on CIFAR-10, applying $\ell_1$ criterion-based pruning across various groups and sparsity levels. The source and target distributions for the optimal transport (OT) setting were chosen as either uniform or importance-informed. To assess the significance of distribution choices, we computed the p-value using the Kruskal-Wallis H-test (Kruskal & Wallis, 1952) for each group-sparsity combination. This test evaluates whether the population medians of all distributions are equal, serving as a non-parametric alternative to ANOVA (Heiberger & Neuwirth, 2009).

The resulting p-values for each group-sparsity pair are visualized in Figure 19. We consider p-values less than or equal to 0.05 as statistically significant. Interestingly, the majority of cases do not exhibit a significant difference. However, for groups 10 and 11, with sparsities ranging from 10% to 50%, notable and statistically significant differences between the distributions emerge.

In Table 3, we focus on Group 10 and 11 to discern nuances in performance. The optimal distribution choices are highlighted in green for the best-performing and in red for the least effective. Notably, the uTuS (uniform target, uniform source) and iTuS (importance-informed target, uniform source) configurations appear superior, while uTiS (uniform target, importance-informed source) performs less favorably. However, it is crucial to emphasize that the variations in accuracy are subtle, seldom exceeding one percentage point.

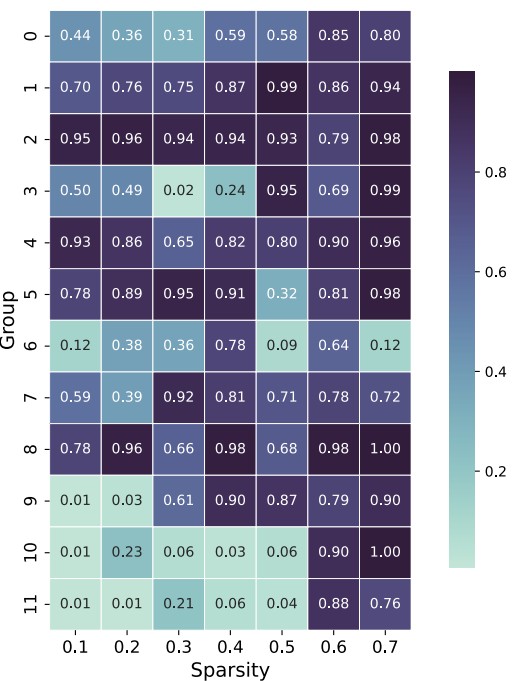

Figure 19: Kruskal-Wallis H-test (Kruskal & Wallis, 1952) p-value between uniform and importance-informed source and target distributions, for different group and sparsity pairs. Model: ResNet18. Dataset: CIFAR-10. Pruning criteria: $\ell_1$.

In light of these findings, we deduce that the selection between uniform or importance-informed distributions for both source and target in the optimal transport (OT) context lacks statistically significant impact in the majority of cases. Furthermore, in instances where statistical significance is observed, the differences remain marginal.

**Remark:** As mentioned before, the pruning criterion used in this study is based on the $\ell_1$-norm. Evidently, this necessarily affects the importance-informed distribution. Thus, it might be possible that some other, more meaningful importance criterion could yield improvements for the importance-informed distributions. However, in light of the results we expect the differences to be more or less marginal.

Table 3: Data-free results for different source and target distributions. uTuS: Uniform target, uniform source. uTis: Uniform target, importance-informed source. iTuS: Importance-informed target, uniform source. iTis: importance-informed target, importance-informed source. Model: ResNet18. Dataset: CIFAR-100. Criterion: $\ell_1$.

| Group | Sparsity (%) | uTuS (%) | uTiS (%) | iTuS (%) | iTiS (%) |
|---|---|---|---|---|---|
| Group 10 | 10 | 94.84 | 94.49 | 94.1 | 94.84 |
| | 20 | 94.6 | 94.31 | 94.49 | 94.55 |
| | 30 | 94.4 | 93.98 | 94.51 | 94.36 |
| | 40 | 94.17 | 93.76 | 94.48 | 94.15 |
| | 50 | 93.3 | 92.86 | 93.81 | 93.24 |
| Group 11 | 10 | 94.83 | 94.48 | 91.89 | 94.78 |
| | 20 | 94.72 | 94.32 | 93.49 | 94.62 |
| | 30 | 94.63 | 94.18 | 94.29 | 94.49 |
| | 40 | 94.37 | 93.97 | 94.56 | 94.15 |
| | 50 | 93.39 | 92.91 | 94.31 | 93.41 |

### C.4 AGNOSTICISM TO IMPORTANCE METRICS

As we have alluded to in Section 3, we argue that the performance improvements of Intra-Fusion are agnostic with respect to the choice of the importance metric. That is, irregardless of the expressiveness of the importance metric, Intra-Fusion is able to achieve a significant gain in accuracy. In order to further highlight this, we compare how Intra-Fusion performs when it is given random scores drawn from a uniform distribution as an importance metric.

As can be seen Table 4, Intra-Fusion still achieves significant increases in accuracy even when only random scores are available. We argue that the superiority of Intra-Fusion is inherent to the integration of the less important neuron pairings in the compressed network. Hence, making it truly agnostic to the importance metric used.

Table 4: Data-free results for ResNet50 on ImageNet. Group 0-10. Random pruning.

| Group | Sparsity (%) | Random (%) | IF (Random)(%) | $\delta$(%) |
|---|---|---|---|---|
| Group 0 | 10 | 72.82 | **75.07** | +2.25 |
| | 20 | 67.97 | **73.96** | +5.99 |
| | 30 | 60.73 | **72.35** | +11.62 |
| | 40 | 53.94 | **70.18** | +16.24 |
| Group 1 | 10 | 75.2 | 75.49 | +0.29 |
| | 20 | 74.19 | 74.62 | +0.43 |
| | 30 | 72.61 | **73.56** | +0.95 |
| | 40 | 70.41 | **72.63** | +2.22 |
| Group 2 | 10 | 74.7 | **75.41** | +0.71 |
| | 20 | 73.33 | **74.55** | +1.22 |
| | 30 | 71.41 | **73.26** | +1.85 |
| | 40 | 69.5 | **71.46** | +1.96 |
| Group 3 | 10 | 75.06 | 75.45 | +0.39 |
| | 20 | 73.98 | **75.02** | +1.04 |
| | 30 | 73.47 | **74.21** | +0.74 |
| | 40 | 71.8 | **73.13** | +1.33 |
| Group 4 | 10 | 75.02 | **75.54** | +0.52 |
| | 20 | 74.09 | **75.09** | +1.0 |
| | 30 | 73.1 | **74.71** | +1.61 |
| | 40 | 72.49 | **73.49** | +1.0 |
| Group 5 | 10 | 75.03 | 75.42 | +0.39 |
| | 20 | 73.48 | **75.0** | +1.52 |
| | 30 | 71.39 | **74.16** | +2.77 |
| | 40 | 70.07 | **73.18** | +3.11 |
| Group 6 | 10 | 74.63 | **75.4** | +0.77 |
| | 20 | 72.44 | **75.16** | +2.72 |
| | 30 | 70.73 | **74.0** | +3.27 |
| | 40 | 69.83 | **72.75** | +2.92 |
| Group 7 | 10 | 72.92 | **74.93** | +2.01 |
| | 20 | 68.3 | **73.14** | +4.84 |
| | 30 | 58.97 | **70.27** | +11.3 |
| | 40 | 52.51 | **65.07** | +12.56 |
| Group 8 | 10 | 75.58 | 75.89 | +0.31 |
| | 20 | 74.79 | **75.56** | +0.77 |
| | 30 | 74.82 | **75.37** | +0.55 |
| | 40 | 74.26 | **74.81** | +0.55 |
| Group 9 | 10 | 75.56 | 75.79 | +0.23 |
| | 20 | 75.27 | 75.61 | +0.34 |
| | 30 | 74.84 | **75.53** | +0.69 |
| | 40 | 74.68 | **75.18** | +0.5 |
| Group 10 | 10 | 75.68 | 75.8 | +0.12 |
| | 20 | 75.43 | 75.61 | +0.18 |
| | 30 | 74.98 | **75.5** | +0.52 |
| | 40 | 74.57 | 74.89 | +0.32 |

# D    APPLICATION BEYOND PRUNING: FACTORIZING MODEL TRAINING

## D.1    RUNTIME COMPARISON OF SPLIT-DATA AND WHOLE-DATA APPROACH

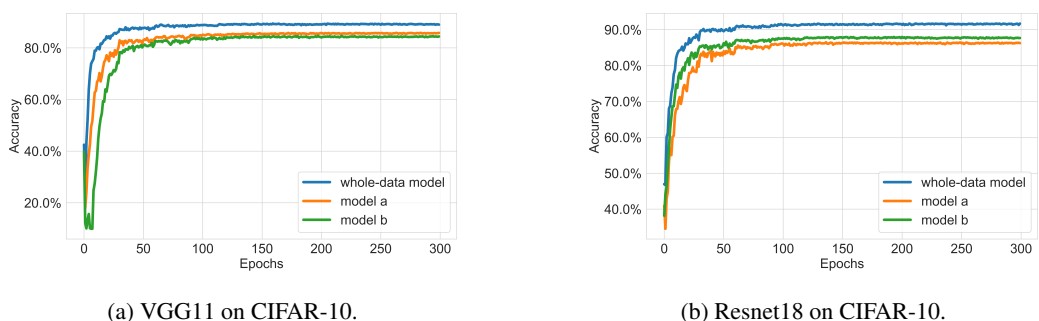

(a) VGG11 on CIFAR-10.                                    (b) Resnet18 on CIFAR-10.

Figure 20: Training convergence speed: comparing whole-data model with the models trained on half the data.

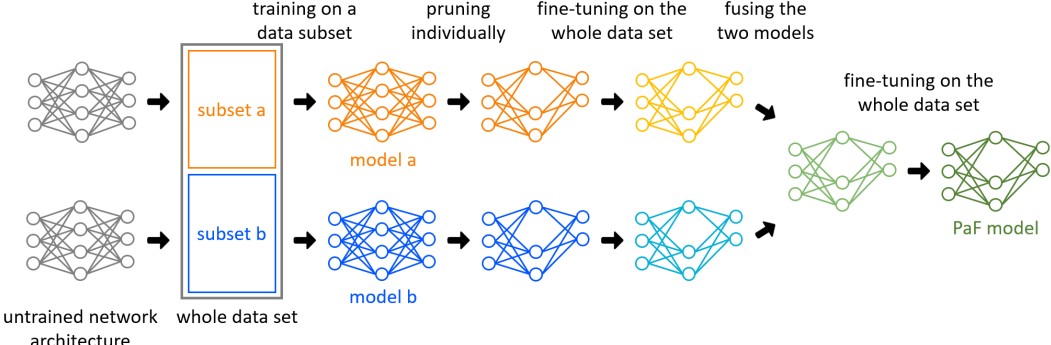

Figure 21: The PaF approach.

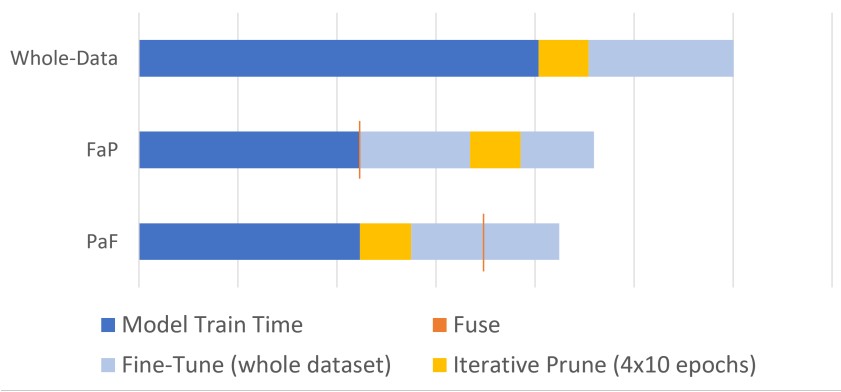

Figure 22: VGG11-bn on CIFAR-10. Relative runtimes for different approaches on an Nvidia-Gpu. Fuse time is so short in comparison to other steps that an extra marking was added for clarity. Speedup of the model train time: 1.81. Overall speedups: 1.31 (FaP), 1.42 (PaF). Further speedups are possible by optimizing the number of fine-tuning epochs at different stages. For models and datasets where the time for model training $T_1$ is much greater than time for fine-tuning $T_2$ ($T_1 \gg T_2$), speedups would show to be much more significant.

### D.2 PERFORMANCE COMPARISON: AFTER CONVERGENCE

To investigate the performance potential of the two approaches, in Figure 23 we show the performance of the PaF and FaP model when the fine-tuning at the intermediate steps is done until convergence. Although there is no clear performance equivalency between the classic whole-data model and the "factorized" model training (FaP and PaF), it is evident that depending on the model architecture and training dynamics the factorization has the potential to even outperform the standard approach. All experiments are done with an iterative pruning approach to recover a stronger performance at higher sparsities. Since data is available for fine-tuning and due to its superior performance, we use activation-based fusion with a sample size of 200.

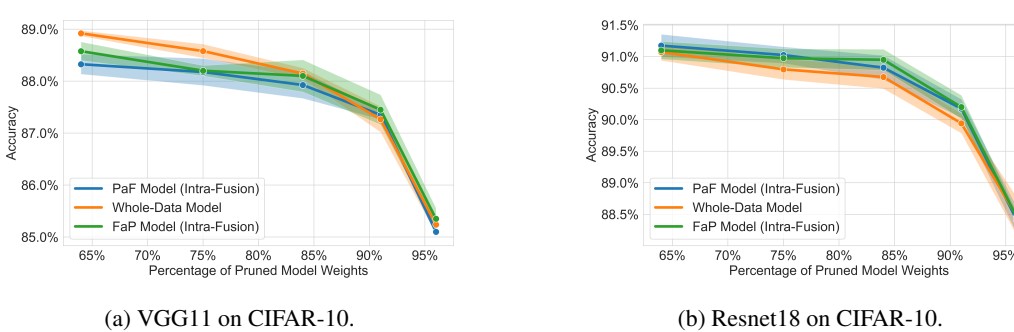

(a) VGG11 on CIFAR-10.  (b) Resnet18 on CIFAR-10.

Figure 23: Performance of PaF and FaP compared to the whole-data model across different sparsities. Here we use Intra-Fusion for pruning in PaF and FaP. An iterative pruning approach is chosen for all three models: 4 steps of each 10 epochs. PaF and FaP are fine-tuned for additional 80 epochs after the iterative pruning and after fusion. The whole-data model is fine-tuned for another 2*80=160 epochs after the iterative pruning. Experiments are done across four different seeds.

For reference, we also compare the performance of the PaF and FaP models using regular pruning (instead of Intra-Fusion) in Figure 24.

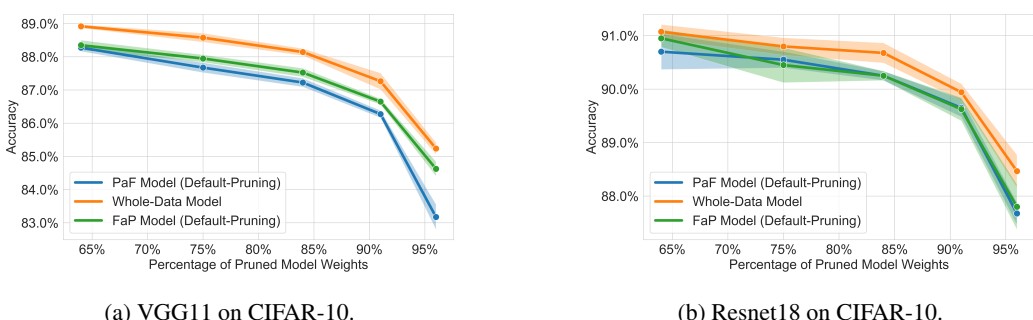

(a) VGG11 on CIFAR-10.  (b) Resnet18 on CIFAR-10.

Figure 24: Performance of PaF and FaP compared to the whole-data model across different sparsities. Here we use regular pruning (instead of Intra-Fusion) for PaF and FaP. An iterative pruning approach is chosen for all three models: 4 steps of each 10 epochs. PaF and FaP are trained for additional 80 epochs after the iterative pruning and after fusion. The whole-data model is trained for another 2*80=160 epochs after the iterative pruning.

### D.3 Performance Comparison: Varying Fine-Tuning

Since the performance of the PaF approach depends on the amount of fine-tuning that is available at the intermediate steps, we also explore how the performance difference to the whole-data model develops with varying amounts of fine-tuning. In this setting, the PaF and FaP models gain a theoreticlal 2x speedup in the training process and take the same time in the post-processing as the whole-data model. In Figure 25 (using Intra-Fusion) and Figure 26 (using conventional pruning) for multiple sparsities, we vary the amount of retraining that is available to the models. This means here we do not get the converged performance of PaF and Fap (only converged at 80 fine-tuning epochs).

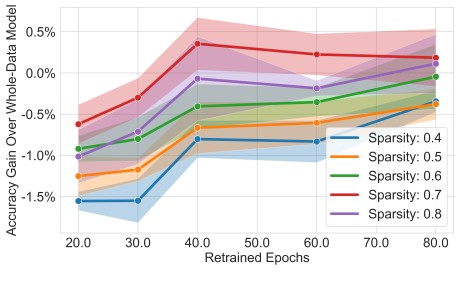

(a) VGG11 on CIFAR-10.

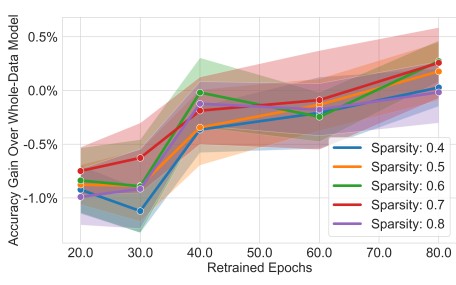

(b) Resnet18 on CIFAR-10.

Figure 25: Comparing the development of performance difference of PaF and the whole-data model when varying the total amount of retraining that is done. PaF, FaP and the whole data model always get the same amount of retraining. Here Intra-Fusion is used in the context of PaF and FaP. Sparsity here is the node sparsity.

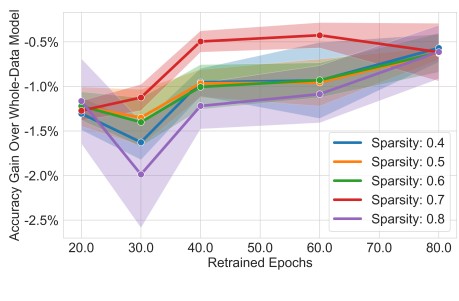

(a) VGG11 on CIFAR-10.

(b) Resnet18 on CIFAR-10.

Figure 26: Comparing the development of performance difference of PaF and the whole-data model when varying the total amount of retraining that is done. PaF, FaP and the whole data model always get the same amount of retraining. Here regular pruning (instead of Intra-Fusion) is used in the context of PaF and FaP. Sparsity here is the node sparsity.

### D.4 k-Fold Split-Data

As an alternative, to simply splitting the dataset into two and using each subset to train a model (that's what we have done so far), we can generalize to a k-fold style approach.

**Generating Additional Trainingsets.** Here we split the dataset into $k$ equally sized and distinct subsets $s_p$. We now create training datasets $d_i$ that consist of $k/2$-many of these subsets $s_p$. By choosing $k \bmod 2 = 0$ we ensure that each $d_i$ will end up containing 50% of the original dataset. We then take all possible $\binom{k}{k/2}$-many $d_i$ and individually train models on them. It is obvious that the 50/50 split of the dataset that we considered in the previous split-data experiments can also be interpreted as a k-fold style approach with $k = 2$, yielding $\binom{2}{1} = 2$ different models ("model a" and "model b" in Figure 27a).

**Extending PaF and FaP.** Since the fusion algorithm naturally extends to fusing more than two models (as also presented by (Singh & Jaggi, 2020)), we can now generalize PaF and FaP to combine pruning and fusion of more than two models - leading to a more effective use of the OT-based fusion approach.

**Consequences for Model Training Speedup.** This approach comes with additional requirements for computational resources to enable the parallel training and pruning of the multiple individually trained models. However, besides fusion taking insignificantly longer, it does not require more time than the previously explored split-data approaches and thus yields the same speedup.

### D.4.1 PERFORMANCE COMPARISON ACROSS DIFFERENT K

To explore the potential of the k-fold approach we also evaluated the performance for the next even choice of $k$, namely $k = 4$. This already yields $\binom{4}{2} = 6$ different models that are combined in PaF and FaP (see models "a" to "f" in Figure 27b).

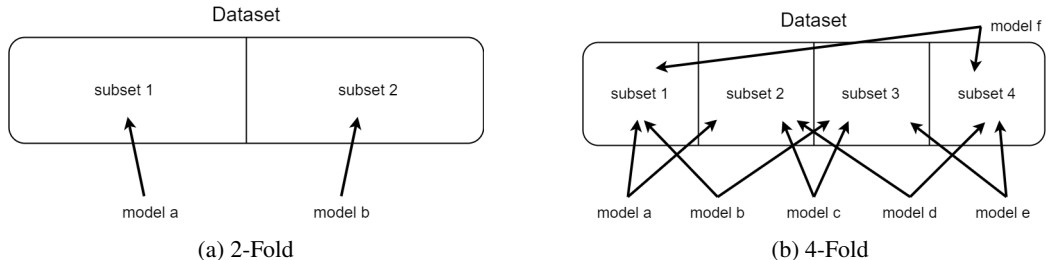

(a) 2-Fold                                                    (b) 4-Fold

Figure 27: Visualization of k-fold data splits at different k.

The performance of PaF and FaP based on the $k = 2$ and $k = 4$ can be compared in Figure 28. For the VGG11-BN we observe performance improvements of up to 1%. Here is important to note that across all measured sparsities the 4-Fold approach always outperforms the 2-Fold approach and yields a very competitive performance when compared to the benchmark "Whole Data Model". For the Resnet18 we seem to not make any improvements in performance by extending from two to six combined models.

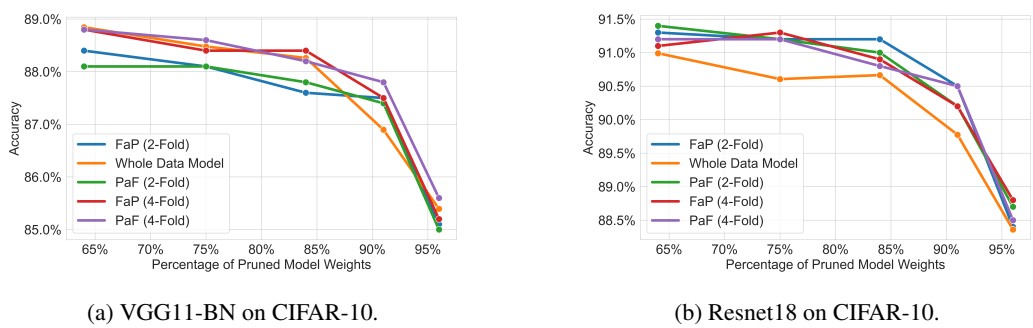

(a) VGG11-BN on CIFAR-10.                    (b) Resnet18 on CIFAR-10.

Figure 28: Performance comparison of k-fold data splits at different k.

### D.5 EXTENSIONS FOR FUTURE WORK

We leave it for further research to explore less drastic splits of the dataset. We believe that this will lead to a better fine-tuning/accuracy trade-off - especially at lower sparsities. For example, the dataset could be split into overlapping sets that make up 60% or 70% of the original dataset.

## D.6 Performance of Models Used

To also get a feeling for how uncompetitive the performance of the individual split-data models is (since they each were only trained on one half of the data) before deploying our PaF and FaP approach and fine-tuning on the whole dataset we include the model performance figures (across different seeds) in Tables 5 and 7. For each seed a different split of the dataset is generated which the split-data models are trained upon

Table 5: Performance of the used 2-Fold split-data models.

| Model | Training Data | Seed | Accuracy (%) |
|---|---|---|---|
| VGG11-BN | 1. data subset | A | 85.74 |
| | | B | 85.54 |
| | | C | 85.80 |
| | | D | 85.33 |
| | 2. data subset | A | 84.50 |
| | | B | 85.32 |
| | | C | 84.99 |
| | | D | 86.09 |
| Resnet18 | 1. data subset | A | 88.46 |
| | | B | 87.44 |
| | | C | 87.75 |
| | | D | 88.17 |
| | 2. data subset | A | 87.71 |
| | | B | 88.31 |
| | | C | 87.65 |
| | | D | 88.23 |

Table 6: Performance of the used 4-Fold split-data models. Single seed.

| Model | Training Data | Accuracy (%) |
|---|---|---|
| VGG11-BN | 1. data subset | 85.36 |
| | 2. data subset | 86.00 |
| | 3. data subset | 85.56 |
| | 4. data subset | 85.22 |
| | 5. data subset | 83.96 |
| | 6. data subset | 85.69 |
| Resnet18 | 1. data subset | 87.87 |
| | 2. data subset | 87.67 |
| | 3. data subset | 87.99 |
| | 4. data subset | 87.91 |
| | 5. data subset | 88.37 |
| | 6. data subset | 87.75 |

Table 7: Performance of the used whole-data models.

| Model | Seed | Accuracy (%) |
|---|---|---|
| VGG11-BN | A | 89.28 |
| | B | 89.47 |
| | C | 89.04 |
| | D | 89.21 |
| Resnet18 | A | 91.47 |
| | B | 92.08 |
| | C | 91.36 |
| | D | 91.64 |

# E EMPIRICAL RESULTS

## E.1 TERMINOLOGY

In Table 8, we show how Neuron Sparsity (applied to all groups in the model) translates to Weight Sparsity.

Table 8: Neuron Sparsity to Weight Sparsity Translation.

| Neuron Sparsity | VGG11_bn #Parameters | Resnet18 #Parameters | Weight Sparsity |
|---|---|---|---|
| Original (0%) | 9,753,674 | 11,164,352 | - |
| 40% | 3,505,301 | 4,003,668 | ~64% |
| 50% | 2,441,770 | 2,792,800 | ~75% |
| 60% | 1,551,390 | 1,772,832 | ~84% |
| 70% | 871,988 | 994,402 | ~91% |
| 80% | 388,770 | 442,308 | ~96% |

## E.2 DATA-DRIVEN EXPERIMENTS

Although fine-tuning may not always be the most convenient in all scenarios, it might be possible in others. In any case, it would be interesting to see whether the performance gains delivered by Intra-Fusion standup in the face of fine-tuning or not. Hence, we carry out a similar experiment as before for both VGG11-BN and ResNet18 trained on CIFAR-10; however, this time, fine-tuning after model compression is available.

Table 9: Intra-Fusion vs. default pruning with fine-tuning for $\ell_1$ importance on CIFAR-10.

| Model | Base (%) | Sparsity (%) | Default (%) | Intra-Fusion (%) | Gain. (%) |
|---|---|---|---|---|---|
| VGG11-BN | 89.56% | 64 | 88.95 | **89.19** | +0.24 |
|  |  | 84 | 88.14 | **88.51** | +0.36 |
|  |  | 91 | 87.43 | **88.15** | +0.72 |
|  |  | 96 | 85.21 | **85.89** | +0.68 |
| ResNet18 | 92.17% | 64 | 91.53 | **91.98** | +0.45 |
|  |  | 84 | 91.22 | **91.62** | +0.40 |
|  |  | 91 | 90.41 | **91.37** | +0.96 |
|  |  | 96 | 88.82 | **89.60** | +0.79 |

Table 9 contains our results (averaged over multiple runs) for this setting. We observe that Intra-Fusion obtains a consistent gain of up to 1% test accuracy (with standard deviation of 0.13% for all sparsities), across all the considered sparsity levels. While the gains might not seem as stark, we must remark that here we allowed for a significantly long fine-tuning schedule, and that Intra-Fusion converges faster due to the large initial accuracy gains. It is also important to note that the focus of this paper is on data-free pruning.

To conclude, our consistent gains show that the boost afforded by Intra-Fusion is complementary to that provided via just fine-tuning the pruned model — thereby demonstrating the efficacy of our approach.

## E.3 DATA-FREE EXPERIMENTS

Here, we provide a full list of results for the data-free experiments. The indices again indicate how close a group is to the output of the model, i.e. Group 0 is the last group of the network. Moreover, we color-code every entry where the absolute difference between the default and Intra-Fused model is greater than 0.5%.

Table 10: Data-free results for VGG11-BN on CIFAR-10. Group 0-5.

| Group | Sparsity (%) | $\ell_1$(%) | IF ($\ell_1$)(%) | $\delta$(%) | Taylor(%) | IF (Taylor)(%) | $\delta$(%) | LAMP(%) | IF (LAMP)(%) | $\delta$(%) |
|---|---|---|---|---|---|---|---|---|---|---|
| Group 0 | 10 | 89.53 | 89.47 | -0.06 | 89.47 | 89.46 | -0.01 | 89.53 | 89.52 | -0.01 |
| | 20 | 89.38 | 89.41 | +0.03 | 89.40 | 89.55 | +0.15 | 89.54 | 89.49 | -0.05 |
| | 30 | 89.30 | 89.31 | +0.01 | 89.27 | 89.46 | +0.19 | 89.44 | 89.56 | +0.12 |
| | 40 | 88.86 | 89.21 | +0.35 | 88.67 | **89.45** | +0.78 | 89.14 | 89.50 | +0.36 |
| | 50 | 87.91 | **89.07** | +1.17 | 87.55 | **89.39** | +1.84 | 88.85 | **89.46** | +0.60 |
| | 60 | 87.42 | **89.36** | +1.94 | 85.40 | **89.21** | +3.81 | 88.88 | **89.41** | +0.52 |
| | 70 | 86.78 | **89.29** | +2.51 | 80.82 | **88.93** | +8.11 | 88.43 | **89.25** | +0.82 |
| Group 1 | 10 | 89.31 | 89.52 | +0.21 | 89.45 | 89.53 | +0.08 | 89.22 | 89.44 | +0.22 |
| | 20 | 88.99 | 89.47 | +0.47 | 89.16 | 89.46 | +0.30 | 88.81 | **89.41** | +0.60 |
| | 30 | 88.74 | 89.15 | +0.42 | 88.76 | 89.24 | +0.48 | 87.25 | **89.28** | +2.03 |
| | 40 | 87.68 | **88.84** | +1.17 | 88.51 | 88.92 | +0.42 | 84.88 | **88.84** | +3.97 |
| | 50 | 85.44 | **88.06** | +2.62 | 87.49 | **88.23** | +0.74 | 81.14 | **88.67** | +7.53 |
| | 60 | 77.81 | **87.58** | +9.77 | 83.55 | **87.53** | +3.98 | 68.12 | **88.25** | +20.13 |
| | 70 | 60.63 | **86.82** | +26.19 | 68.88 | **86.60** | +17.72 | 44.23 | **88.02** | +43.79 |
| Group 2 | 10 | 87.67 | **89.41** | +1.74 | 89.44 | 89.37 | -0.07 | 88.82 | **89.50** | +0.68 |
| | 20 | 85.17 | **88.92** | +3.76 | 88.99 | 89.00 | +0.01 | 86.89 | **89.21** | +2.32 |
| | 30 | 80.41 | **88.26** | +7.85 | 87.34 | **88.46** | +1.12 | 82.80 | **88.65** | +5.84 |
| | 40 | 76.07 | **87.18** | +11.12 | 84.61 | **87.36** | +2.75 | 77.29 | **87.98** | +10.69 |
| | 50 | 68.41 | **85.37** | +16.96 | 78.94 | **85.11** | +6.17 | 70.82 | **86.55** | +15.73 |
| | 60 | 60.10 | **83.51** | +23.42 | 74.10 | **83.70** | +9.60 | 48.04 | **84.21** | +36.16 |
| | 70 | 37.89 | **80.11** | +42.23 | 65.62 | **80.86** | +15.25 | 28.00 | **80.80** | +52.81 |
| Group 3 | 10 | 89.11 | 89.28 | +0.17 | 89.30 | 89.35 | +0.05 | 89.41 | 89.42 | +0.01 |
| | 20 | 88.49 | 88.84 | +0.35 | 89.06 | 88.76 | -0.31 | 88.70 | **89.30** | +0.60 |
| | 30 | 86.76 | **87.79** | +1.03 | 88.50 | 88.06 | -0.44 | 87.79 | **88.99** | +1.21 |
| | 40 | 85.56 | **87.45** | +1.89 | 87.47 | 87.12 | -0.35 | 86.73 | **88.47** | +1.74 |
| | 50 | 79.49 | **86.68** | +7.19 | 85.52 | **86.06** | +0.53 | 83.51 | **87.93** | +4.41 |
| | 60 | 72.23 | **85.58** | +13.35 | 82.58 | **86.15** | +3.57 | 76.07 | **86.94** | +10.87 |
| | 70 | 54.86 | **84.10** | +29.24 | 71.07 | **85.10** | +14.02 | 59.39 | **85.58** | +26.19 |
| Group 4 | 10 | 89.13 | 89.20 | +0.07 | 89.16 | 89.16 | +0.00 | 88.55 | **89.38** | +0.83 |
| | 20 | 88.79 | 88.45 | -0.34 | **88.29** | 87.59 | -0.70 | 84.29 | **88.40** | +4.11 |
| | 30 | 87.06 | 86.64 | -0.43 | 86.31 | 86.74 | +0.43 | 82.46 | **88.04** | +5.59 |
| | 40 | 82.33 | **85.04** | +2.71 | 81.69 | **84.52** | +2.84 | 80.17 | **87.10** | +6.93 |
| | 50 | 65.26 | **80.29** | +15.03 | 73.31 | **82.81** | +9.50 | 74.15 | **83.98** | +9.83 |
| | 60 | 51.82 | **80.24** | +28.42 | 57.40 | **80.32** | +22.92 | 68.52 | **82.74** | +14.22 |
| | 70 | 47.16 | **79.02** | +31.85 | 37.24 | **76.56** | +39.32 | 46.77 | **77.39** | +30.63 |
| Group 5 | 10 | 89.17 | 88.85 | -0.32 | 89.27 | 89.00 | -0.27 | 87.97 | **89.25** | +1.28 |
| | 20 | 87.76 | 87.65 | -0.11 | 87.86 | 88.03 | +0.18 | 85.45 | **88.59** | +3.13 |
| | 30 | 85.67 | **86.56** | +0.89 | 86.48 | **87.58** | +1.10 | 84.58 | **88.18** | +3.60 |
| | 40 | 83.49 | **84.92** | +1.42 | 83.72 | **86.05** | +2.32 | 82.52 | **85.87** | +3.35 |
| | 50 | **79.45** | 78.75 | -0.70 | **81.26** | 80.57 | -0.69 | 76.41 | **82.30** | +5.88 |
| | 60 | 74.63 | **75.26** | +0.62 | 72.66 | **77.44** | +4.79 | 67.12 | **78.68** | +11.56 |
| | 70 | 62.81 | **67.15** | +4.34 | 59.04 | **68.87** | +9.83 | 49.90 | **69.93** | +20.03 |

Table 11: Data-free results for VGG11-BN on CIFAR-10. Group 6-7.

| Group | Sparsity (%) | $\ell_1$(%) | IF ($\ell_1$)(%) | $\delta$(%) | Taylor(%) | IF (Taylor)(%) | $\delta$(%) | LAMP(%) | IF (LAMP)(%) | $\delta$(%) |
|---|---|---|---|---|---|---|---|---|---|---|
| | 10 | 88.59 | 88.76 | +0.17 | 88.88 | 88.39 | -0.49 | 87.38 | **89.00** | +1.62 |
| | 20 | 87.05 | **87.78** | +0.73 | 87.32 | 87.69 | +0.37 | 83.39 | **88.26** | +4.88 |
| Group 6 | 30 | 83.59 | **86.56** | +2.97 | 80.06 | **84.90** | +4.84 | 79.21 | **86.42** | +7.21 |
| | 40 | 78.80 | **82.72** | +3.93 | 57.45 | **82.03** | +24.58 | 59.87 | **84.33** | +24.46 |
| | 50 | 59.99 | **71.37** | +11.38 | 48.39 | **70.92** | +22.53 | 40.62 | **76.15** | +35.52 |
| | 60 | 35.62 | **64.56** | +28.94 | 32.46 | **66.00** | +33.54 | 28.70 | **66.78** | +38.08 |
| | 70 | 16.66 | **51.69** | +35.03 | 21.35 | **51.91** | +30.56 | 20.41 | **49.87** | +29.46 |
| | 10 | **89.10** | 88.00 | -1.10 | **89.41** | 86.48 | -2.93 | 84.84 | **88.83** | +3.99 |
| | 20 | **88.71** | 86.51 | -2.20 | **89.17** | 84.48 | -4.69 | 57.63 | **84.20** | +26.56 |
| Group 7 | 30 | **88.22** | 79.10 | -9.12 | **88.19** | 78.35 | -9.84 | 50.06 | **83.44** | +33.38 |
| | 40 | **87.19** | 68.28 | -18.92 | **87.23** | 70.19 | -17.04 | 41.91 | **80.88** | +38.97 |
| | 50 | **84.23** | 63.05 | -21.17 | **83.84** | 47.18 | -36.66 | 34.23 | **75.74** | +41.52 |
| | 60 | **65.06** | 57.42 | -7.64 | **75.65** | 47.05 | -28.60 | 28.81 | **69.65** | +40.84 |
| | 70 | 48.07 | **49.67** | +1.60 | **48.08** | 41.42 | -6.67 | 23.30 | **60.77** | +37.47 |

Table 12: Data-free results for a ResNet18 on CIFAR-10. Group 0-5.

| Group | Sparsity (%) | $\ell_1$(%) | IF ($\ell_1$)(%) | $\delta$(%) | Taylor(%) | IF (Taylor)(%) | $\delta$(%) | LAMP(%) | IF (LAMP)(%) | $\delta$(%) |
|---|---|---|---|---|---|---|---|---|---|---|
| Group 0 | 10 | 94.89 | 94.91 | +0.02 | 94.88 | 94.84 | -0.04 | 94.71 | 94.70 | -0.01 |
| | 20 | 94.81 | 94.82 | +0.01 | 94.88 | 94.78 | -0.03 | 94.46 | 94.64 | +0.17 |
| | 30 | 94.75 | 94.79 | +0.04 | 94.77 | 94.66 | -0.11 | 94.23 | 94.55 | +0.32 |
| | 40 | 94.50 | 94.68 | +0.18 | 94.59 | 94.49 | -0.10 | 93.56 | **94.56** | +1.00 |
| | 50 | 94.35 | 94.56 | +0.21 | 94.22 | 94.43 | +0.21 | 92.83 | **94.51** | +1.68 |
| | 60 | 93.81 | **94.39** | +0.58 | 92.37 | **94.10** | +1.73 | 89.54 | **94.16** | +4.62 |
| | 70 | 92.90 | **93.98** | +1.08 | 87.81 | **93.82** | +6.01 | 74.56 | **93.82** | +19.26 |
| Group 1 | 10 | 94.86 | 94.93 | +0.07 | 94.90 | 94.85 | -0.05 | 94.81 | 94.76 | -0.05 |
| | 20 | 94.88 | 94.88 | +0.00 | 94.88 | 94.87 | -0.01 | 94.80 | 94.80 | +0.00 |
| | 30 | 94.86 | 94.89 | +0.04 | 94.72 | 94.88 | +0.16 | 94.70 | 94.83 | +0.13 |
| | 40 | 94.84 | 94.82 | -0.02 | 94.65 | 94.89 | +0.24 | 94.64 | 94.74 | +0.09 |
| | 50 | 94.72 | 94.86 | +0.14 | 94.36 | 94.86 | +0.49 | 94.38 | 94.78 | +0.40 |
| | 60 | 94.61 | 94.85 | +0.24 | 94.18 | **94.80** | +0.62 | 94.05 | **94.70** | +0.65 |
| | 70 | 94.09 | **94.83** | +0.74 | 93.13 | **94.62** | +1.49 | 91.54 | **94.65** | +3.11 |
| Group 2 | 10 | 94.87 | 94.91 | +0.04 | 94.86 | 94.85 | -0.01 | 94.71 | 94.84 | +0.13 |
| | 20 | 94.75 | 94.87 | +0.12 | 94.87 | 94.75 | -0.12 | 94.67 | 94.79 | +0.11 |
| | 30 | 94.58 | 94.86 | +0.28 | 94.72 | 94.75 | +0.03 | 94.49 | 94.73 | +0.25 |
| | 40 | 94.34 | 94.82 | +0.48 | 94.48 | 94.74 | +0.26 | 94.19 | **94.70** | +0.51 |
| | 50 | 94.10 | **94.76** | +0.66 | 94.06 | **94.59** | +0.53 | 93.77 | **94.57** | +0.80 |
| | 60 | 93.50 | **94.61** | +1.11 | 93.67 | **94.30** | +0.63 | 93.35 | **94.59** | +1.24 |
| | 70 | 92.41 | **94.41** | +2.00 | 92.90 | **94.07** | +1.17 | 92.64 | **94.33** | +1.69 |
| Group 3 | 10 | 94.25 | 94.56 | +0.31 | 94.20 | 94.52 | +0.32 | 93.96 | 94.46 | +0.49 |
| | 20 | 93.22 | **94.09** | +0.87 | 92.95 | **93.93** | +0.98 | 93.17 | **93.99** | +0.82 |
| | 30 | 91.33 | **93.59** | +2.26 | 91.93 | **93.43** | +1.50 | 91.85 | **93.44** | +1.59 |
| | 40 | 87.75 | **92.46** | +4.71 | 89.17 | **92.58** | +3.41 | 90.06 | **92.61** | +2.56 |
| | 50 | 81.91 | **91.25** | +9.34 | 83.69 | **91.67** | +7.98 | 85.43 | **90.45** | +5.01 |
| | 60 | 75.73 | **88.32** | +12.59 | 76.30 | **88.42** | +12.12 | 80.52 | **88.75** | +8.23 |
| | 70 | 58.93 | **80.70** | +21.77 | 57.86 | **79.97** | +22.11 | 62.23 | **80.91** | +18.68 |
| Group 4 | 10 | 94.85 | 94.88 | +0.03 | **94.78** | 10.10 | -84.68 | 94.60 | 94.65 | +0.05 |
| | 20 | 94.69 | 94.75 | +0.06 | **94.44** | 10.10 | -84.35 | 94.27 | 94.49 | +0.22 |
| | 30 | 94.43 | 94.54 | +0.10 | **94.17** | 10.10 | -84.07 | 93.90 | 94.39 | +0.49 |
| | 40 | 94.18 | 94.30 | +0.12 | **93.91** | 10.10 | -83.81 | 93.43 | **94.13** | +0.69 |
| | 50 | 93.71 | 94.02 | +0.31 | **93.57** | 10.10 | -83.48 | 92.87 | **93.92** | +1.05 |
| | 60 | 93.11 | **93.76** | +0.65 | 92.67 | 10.10 | -82.58 | 92.38 | **93.38** | +1.00 |
| | 70 | 91.96 | **92.70** | +0.74 | 91.81 | 10.10 | -81.71 | 91.65 | **92.62** | +0.97 |
| Group 5 | 10 | 94.73 | 94.84 | +0.11 | 94.52 | 94.77 | +0.25 | 94.38 | 94.64 | +0.26 |
| | 20 | 94.37 | 94.67 | +0.31 | 94.04 | **94.63** | +0.59 | 93.95 | **94.59** | +0.64 |
| | 30 | 93.84 | **94.58** | +0.74 | 93.68 | **94.50** | +0.82 | 93.61 | **94.33** | +0.72 |
| | 40 | 93.03 | **94.40** | +1.37 | 92.98 | **94.30** | +1.33 | 92.77 | **94.09** | +1.33 |
| | 50 | 91.21 | **93.66** | +2.45 | 91.24 | **93.62** | +2.38 | 91.40 | **93.40** | +2.00 |
| | 60 | 88.30 | **93.10** | +4.80 | 88.93 | **93.14** | +4.20 | 89.34 | **92.85** | +3.51 |
| | 70 | 82.40 | **91.05** | +8.65 | 84.31 | **91.23** | +6.92 | 85.76 | **91.05** | +5.29 |

Table 13: Data-free results for a ResNet18 on CIFAR-10. Group 6-10.

| Group | Sparsity (%) | $\ell_1$(%) | IF ($\ell_1$)(%) | $\delta$(%) | Taylor(%) | IF (Taylor)(%) | $\delta$(%) | LAMP(%) | IF (LAMP)(%) | $\delta$(%) |
|---|---|---|---|---|---|---|---|---|---|---|
| Group 6 | 10 | 93.46 | **94.58** | +1.12 | 94.20 | 94.60 | +0.41 | 93.80 | **94.54** | +0.74 |
| | 20 | 89.98 | **94.20** | +4.22 | 92.44 | **94.42** | +1.98 | 91.17 | **94.04** | +2.87 |
| | 30 | 85.69 | **93.47** | +7.78 | 90.64 | **93.70** | +3.06 | 87.71 | **93.69** | +5.98 |
| | 40 | 78.73 | **92.27** | +13.54 | 85.92 | **93.60** | +7.68 | 78.96 | **92.48** | +13.52 |
| | 50 | 67.09 | **86.44** | +19.35 | 79.16 | **90.85** | +11.69 | 61.25 | **85.41** | +24.16 |
| | 60 | 50.30 | **83.08** | +32.78 | 70.92 | **87.90** | +16.98 | 38.66 | **83.69** | +45.03 |
| | 70 | 29.14 | **64.10** | +34.96 | 49.26 | **61.44** | +12.18 | 23.06 | **61.76** | +38.70 |
| Group 7 | 10 | 94.89 | 94.92 | +0.02 | **94.89** | 10.10 | -84.79 | 94.70 | 94.78 | +0.08 |
| | 20 | 94.74 | 94.88 | +0.15 | **94.73** | 10.10 | -84.63 | 94.33 | 94.64 | +0.31 |
| | 30 | 94.71 | 94.71 | -0.00 | **94.75** | 10.10 | -84.65 | 93.87 | **94.50** | +0.62 |
| | 40 | 94.57 | 94.59 | +0.02 | **94.54** | 10.10 | -84.44 | 93.42 | **94.55** | +1.13 |
| | 50 | 94.37 | 94.32 | -0.05 | **94.33** | 10.10 | -84.24 | 93.10 | **94.29** | +1.19 |
| | 60 | 94.02 | 94.10 | +0.07 | **94.11** | 10.10 | -84.01 | 92.10 | **94.32** | +2.23 |
| | 70 | 93.25 | **93.77** | +0.52 | 93.23 | 10.10 | -83.13 | 90.27 | **93.77** | +3.50 |
| Group 8 | 10 | 94.80 | 94.87 | +0.07 | 94.71 | 94.71 | +0.00 | 94.21 | **94.74** | +0.53 |
| | 20 | 94.48 | 94.76 | +0.28 | 94.47 | 94.64 | +0.17 | 93.44 | **94.49** | +1.04 |
| | 30 | 93.97 | **94.59** | +0.62 | 94.28 | 94.45 | +0.17 | 90.71 | **93.82** | +3.11 |
| | 40 | 93.02 | **93.74** | +0.72 | 93.32 | **93.95** | +0.62 | 87.82 | **93.04** | +5.23 |
| | 50 | 91.03 | 91.42 | +0.39 | 91.47 | **92.28** | +0.81 | 82.35 | **88.71** | +6.36 |
| | 60 | 86.78 | **88.72** | +1.93 | 86.46 | **91.16** | +4.70 | 69.53 | **86.15** | +16.62 |
| | 70 | 77.14 | **79.06** | +1.91 | 75.05 | **86.84** | +11.79 | 49.80 | **70.09** | +20.29 |
| Group 9 | 10 | 94.26 | 94.38 | +0.11 | 94.56 | 94.35 | -0.21 | 93.03 | **93.89** | +0.86 |
| | 20 | 91.28 | **93.40** | +2.13 | 94.04 | 93.87 | -0.17 | 84.66 | **92.62** | +7.97 |
| | 30 | 82.56 | **91.57** | +9.01 | 91.48 | **92.61** | +1.14 | 57.60 | **86.02** | +28.42 |
| | 40 | 70.47 | **87.74** | +17.27 | 85.47 | **93.21** | +7.73 | 42.69 | **81.29** | +38.61 |
| | 50 | 57.92 | **71.75** | +13.84 | 79.20 | **89.01** | +9.81 | 16.63 | **74.11** | +57.49 |
| | 60 | 40.01 | **64.03** | +24.02 | 65.89 | **79.90** | +14.00 | 12.52 | **41.70** | +29.18 |
| | 70 | 23.41 | **45.47** | +22.05 | 40.43 | **69.15** | +28.72 | 10.21 | **17.16** | +6.96 |
| Group 10 | 10 | 94.91 | 94.84 | -0.07 | 94.88 | 94.80 | -0.08 | 94.45 | 94.73 | +0.28 |
| | 20 | 94.89 | 94.66 | -0.23 | 94.82 | 94.37 | -0.45 | 93.91 | **94.47** | +0.55 |
| | 30 | 94.83 | 94.42 | -0.41 | **94.87** | 94.22 | -0.64 | 93.23 | **94.45** | +1.22 |
| | 40 | 94.59 | 94.21 | -0.37 | **94.85** | 93.89 | -0.96 | 93.13 | **94.46** | +1.33 |
| | 50 | **94.33** | 93.51 | -0.82 | **94.73** | 93.58 | -1.15 | 90.27 | **93.62** | +3.35 |
| | 60 | **93.61** | 92.69 | -0.92 | **94.22** | 93.09 | -1.14 | 89.17 | **92.41** | +3.23 |
| | 70 | 91.41 | 91.70 | +0.29 | **92.28** | 90.70 | -1.57 | 86.45 | **89.01** | +2.56 |

Table 14: Data-free results on VGG11-BN on CIFAR-100. Group 0-5.

| Group | Sparsity (%) | $\ell_1$(%) | IF ($\ell_1$)(%) | $\delta$(%) | Taylor(%) | IF (Taylor)(%) | $\delta$(%) | LAMP(%) | IF (LAMP)(%) | $\delta$(%) |
|---|---|---|---|---|---|---|---|---|---|---|
| Group 0 | 10 | 66.96 | **67.51** | +0.55 | 65.53 | **66.16** | +0.63 | 65.39 | 65.80 | +0.42 |
| | 20 | 65.75 | **67.02** | +1.27 | 63.23 | **65.92** | +2.69 | 64.53 | **65.41** | +0.88 |
| | 30 | 63.94 | **66.46** | +2.51 | 61.14 | **64.96** | +3.83 | 62.87 | **64.91** | +2.05 |
| | 40 | 62.12 | **65.77** | +3.65 | 57.75 | **64.48** | +6.72 | 60.84 | **63.58** | +2.74 |
| | 50 | 59.88 | **65.09** | +5.21 | 52.46 | **64.02** | +11.56 | 56.55 | **63.03** | +6.49 |
| | 60 | 54.46 | **62.93** | +8.47 | 46.61 | **61.76** | +15.15 | 51.60 | **60.62** | +9.02 |
| | 70 | 46.92 | **60.07** | +13.14 | 38.54 | **59.21** | +20.67 | 44.77 | **57.42** | +12.65 |
| Group 1 | 10 | 66.42 | **67.03** | +0.61 | 64.83 | **65.99** | +1.16 | 65.35 | **65.70** | +0.36 |
| | 20 | 63.43 | **66.70** | +3.27 | 61.88 | **65.38** | +3.50 | 63.42 | **65.06** | +1.64 |
| | 30 | 57.36 | **65.51** | +8.15 | 57.79 | **64.20** | +6.41 | 61.70 | **64.03** | +2.33 |
| | 40 | 47.87 | **63.87** | +16.00 | 52.58 | **62.78** | +10.20 | 58.02 | **62.48** | +4.46 |
| | 50 | 37.97 | **62.39** | +24.42 | 44.30 | **60.96** | +16.65 | 51.99 | **60.85** | +8.86 |
| | 60 | 26.97 | **59.38** | +32.42 | 36.47 | **56.50** | +20.03 | 40.54 | **57.46** | +16.92 |
| | 70 | 17.01 | **52.65** | +35.64 | 26.15 | **51.09** | +24.94 | 25.02 | **51.84** | +26.82 |
| Group 2 | 10 | 65.92 | **67.24** | +1.32 | 64.88 | **66.01** | +1.13 | 65.00 | **66.04** | +1.04 |
| | 20 | 62.49 | **66.74** | +4.25 | 62.10 | **65.22** | +3.12 | 63.53 | **65.18** | +1.65 |
| | 30 | 56.68 | **65.60** | +8.92 | 58.82 | **64.56** | +5.74 | 59.71 | **64.40** | +4.69 |
| | 40 | 50.29 | **64.14** | +13.85 | 54.55 | **63.03** | +8.48 | 54.10 | **63.25** | +9.15 |
| | 50 | 40.53 | **62.68** | +22.15 | 47.24 | **60.31** | +13.07 | 46.82 | **60.58** | +13.77 |
| | 60 | 24.31 | **60.10** | +35.79 | 36.85 | **58.17** | +21.32 | 32.31 | **58.51** | +26.21 |
| | 70 | 13.53 | **54.58** | +41.05 | 19.68 | **53.26** | +33.58 | 17.20 | **53.84** | +36.64 |
| Group 3 | 10 | 66.35 | **67.55** | +1.21 | 65.09 | **66.13** | +1.04 | 65.35 | **66.20** | +0.85 |
| | 20 | 62.89 | **67.00** | +4.11 | 62.39 | **65.48** | +3.09 | 63.42 | **65.64** | +2.23 |
| | 30 | 58.79 | **66.04** | +7.25 | 59.89 | **65.07** | +5.18 | 61.16 | **64.93** | +3.77 |
| | 40 | 50.44 | **65.07** | +14.64 | 54.36 | **64.17** | +9.81 | 57.01 | **63.49** | +6.48 |
| | 50 | 41.80 | **62.12** | +20.32 | 47.59 | **60.57** | +12.98 | 48.83 | **60.54** | +11.71 |
| | 60 | 31.21 | **60.42** | +29.21 | 35.29 | **58.70** | +23.41 | 37.66 | **58.19** | +20.53 |
| | 70 | 20.74 | **53.43** | +32.69 | 22.84 | **52.06** | +29.21 | 20.98 | **52.04** | +31.05 |
| Group 4 | 10 | 65.00 | **67.17** | +2.17 | 63.35 | **65.76** | +2.41 | 63.93 | **65.85** | +1.92 |
| | 20 | 58.71 | **66.12** | +7.41 | 59.33 | **64.94** | +5.62 | 59.32 | **65.04** | +5.73 |
| | 30 | 52.00 | **65.18** | +13.18 | 52.89 | **64.11** | +11.22 | 54.52 | **63.24** | +8.72 |
| | 40 | 37.99 | **63.16** | +25.17 | 45.60 | **61.75** | +16.15 | 46.91 | **61.39** | +14.48 |
| | 50 | 22.63 | **58.89** | +36.26 | 35.16 | **57.87** | +22.72 | 37.08 | **56.27** | +19.19 |
| | 60 | 13.11 | **54.19** | +41.08 | 18.60 | **52.74** | +34.14 | 18.32 | **53.83** | +35.50 |
| | 70 | 5.10 | **45.65** | +40.55 | 7.80 | **46.37** | +38.57 | 7.66 | **45.50** | +37.84 |
| Group 5 | 10 | 66.20 | **67.82** | +1.62 | 65.03 | **66.32** | +1.29 | 63.89 | **66.09** | +2.20 |
| | 20 | 61.23 | **67.29** | +6.05 | 61.19 | **65.58** | +4.38 | 59.71 | **65.57** | +5.85 |
| | 30 | 51.81 | **65.63** | +13.83 | 54.28 | **65.12** | +10.84 | 53.83 | **64.24** | +10.41 |
| | 40 | 37.58 | **63.28** | +25.70 | 42.68 | **63.01** | +20.33 | 44.47 | **62.12** | +17.65 |
| | 50 | 22.12 | **57.66** | +35.54 | 27.74 | **55.85** | +28.12 | 29.66 | **55.66** | +26.00 |
| | 60 | 10.84 | **52.93** | +42.09 | 13.89 | **52.49** | +38.60 | 15.35 | **51.71** | +36.36 |
| | 70 | 5.30 | **41.17** | +35.87 | 8.20 | **43.46** | +35.27 | 8.04 | **44.28** | +36.24 |

Table 15: Data-free results on VGG11-BN on CIFAR-100. Group 6-7.

| Group | Sparsity (%) | $\ell_1$(%) | IF ($\ell_1$)(%) | $\delta$(%) | Taylor(%) | IF (Taylor)(%) | $\delta$(%) | LAMP(%) | IF (LAMP)(%) | $\delta$(%) |
|---|---|---|---|---|---|---|---|---|---|---|
| | 10 | 64.33 | **67.01** | +2.68 | 60.67 | **64.46** | +3.79 | 63.09 | **66.52** | +3.43 |
| | 20 | 54.91 | **65.08** | +10.18 | 48.99 | **63.01** | +14.02 | 55.63 | **64.95** | +9.33 |
| Group 6 | 30 | 39.80 | **61.02** | +21.21 | 36.19 | **59.28** | +23.08 | 42.50 | **62.69** | +20.18 |
| | 40 | 22.04 | **52.87** | +30.82 | 18.26 | **52.03** | +33.77 | 32.98 | **56.08** | +23.10 |
| | 50 | 10.74 | **41.23** | +30.49 | 12.17 | **41.64** | +29.47 | 18.80 | **45.09** | +26.30 |
| | 60 | 5.67 | **30.70** | +25.03 | 8.76 | **34.49** | +25.73 | 7.53 | **38.07** | +30.55 |
| | 70 | 2.98 | **21.34** | +18.36 | 5.06 | **24.19** | +19.13 | 3.33 | **27.16** | +23.82 |
| | 10 | **66.95** | 66.43 | -0.52 | **65.54** | 63.11 | -2.42 | 60.49 | **63.64** | +3.14 |
| | 20 | **65.27** | 61.24 | -4.02 | **64.44** | 59.46 | -4.97 | 49.83 | **60.14** | +10.30 |
| Group 7 | 30 | **60.32** | 57.44 | -2.89 | **58.42** | 47.30 | -11.12 | 36.83 | **54.33** | +17.50 |
| | 40 | 49.00 | **50.27** | +1.27 | **53.25** | 41.28 | -11.98 | 24.64 | **49.68** | +25.04 |
| | 50 | 41.53 | 41.59 | +0.06 | **46.19** | 34.64 | -11.55 | 12.67 | **42.42** | +29.76 |
| | 60 | 24.42 | **31.29** | +6.87 | 18.81 | **27.59** | +8.78 | 6.59 | **33.58** | +27.00 |
| | 70 | 10.26 | **21.77** | +11.51 | 11.83 | **16.07** | +4.24 | 5.71 | **18.73** | +13.02 |

Table 16: Data-free results on ResNet18 on CIFAR-100. Group 0-5.

| Group | Sparsity (%) | $\ell_1$(%) | IF ($\ell_1$)(%) | $\delta$(%) | Taylor(%) | IF (Taylor)(%) | $\delta$(%) | LAMP(%) | IF (LAMP)(%) | $\delta$(%) |
|---|---|---|---|---|---|---|---|---|---|---|
| Group 0 | 10 | 76.01 | 76.16 | +0.15 | 70.30 | **71.71** | +1.40 | 73.63 | **74.98** | +1.35 |
| | 20 | 73.62 | **74.87** | +1.26 | 66.26 | **69.87** | +3.61 | 71.36 | **73.20** | +1.84 |
| | 30 | 70.57 | **72.10** | +1.53 | 60.26 | **66.69** | +6.44 | 66.45 | **71.30** | +4.86 |
| | 40 | 66.24 | **69.31** | +3.08 | 50.57 | **61.49** | +10.92 | 61.44 | **66.90** | +5.46 |
| | 50 | 58.22 | **65.88** | +7.66 | 38.57 | **58.64** | +20.08 | 55.14 | **64.40** | +9.26 |
| | 60 | 48.41 | **55.55** | +7.14 | 28.28 | **46.83** | +18.54 | 43.33 | **54.75** | +11.41 |
| | 70 | 37.10 | **42.58** | +5.48 | 17.46 | **31.77** | +14.31 | 36.69 | **40.57** | +3.88 |
| Group 1 | 10 | 76.38 | 76.85 | +0.46 | 71.48 | **72.13** | +0.65 | 74.89 | 75.38 | +0.48 |
| | 20 | 75.21 | **75.82** | +0.61 | 68.99 | **71.07** | +2.09 | 73.71 | **74.29** | +0.57 |
| | 30 | 74.19 | **74.91** | +0.72 | 66.49 | **69.27** | +2.79 | 72.17 | **72.70** | +0.52 |
| | 40 | 71.00 | **72.69** | +1.68 | 63.24 | **66.55** | +3.31 | 70.01 | **71.85** | +1.84 |
| | 50 | 67.77 | **71.53** | +3.76 | 58.11 | **64.73** | +6.63 | 67.24 | **70.43** | +3.19 |
| | 60 | 62.55 | **64.88** | +2.33 | 48.68 | **55.73** | +7.04 | 62.66 | **63.77** | +1.11 |
| | 70 | **56.11** | 55.59 | -0.52 | 41.71 | **44.67** | +2.96 | **56.22** | 52.40 | -3.82 |
| Group 2 | 10 | 76.69 | **77.22** | +0.52 | 72.09 | 72.49 | +0.40 | 75.03 | 75.34 | +0.31 |
| | 20 | 75.75 | **76.70** | +0.95 | 71.45 | **72.31** | +0.86 | 74.12 | **75.02** | +0.90 |
| | 30 | 74.53 | **76.29** | +1.76 | 70.23 | **72.02** | +1.79 | 72.60 | **74.03** | +1.43 |
| | 40 | 71.95 | **74.84** | +2.89 | 68.41 | **71.06** | +2.65 | 70.97 | **73.22** | +2.25 |
| | 50 | 68.18 | **73.14** | +4.96 | 65.76 | **69.82** | +4.05 | 67.15 | **71.22** | +4.07 |
| | 60 | 62.58 | **70.91** | +8.33 | 62.14 | **68.40** | +6.26 | 62.10 | **68.57** | +6.47 |
| | 70 | 54.84 | **65.63** | +10.80 | 54.65 | **64.66** | +10.01 | 53.48 | **62.10** | +8.62 |
| Group 3 | 10 | 74.07 | **76.45** | +2.38 | 71.61 | **72.36** | +0.75 | 73.36 | **75.18** | +1.82 |
| | 20 | 66.22 | **74.77** | +8.55 | 68.12 | **71.55** | +3.43 | 70.57 | **74.07** | +3.50 |
| | 30 | 53.07 | **72.54** | +19.47 | 62.31 | **70.20** | +7.89 | 63.92 | **71.98** | +8.06 |
| | 40 | 35.92 | **67.97** | +32.05 | 52.52 | **66.56** | +14.04 | 53.54 | **68.42** | +14.88 |
| | 50 | 27.15 | **51.21** | +24.06 | 41.49 | **47.46** | +5.97 | 45.52 | **50.02** | +4.50 |
| | 60 | 15.88 | **40.85** | +24.97 | 32.54 | **39.36** | +6.82 | 29.02 | **39.55** | +10.52 |
| | 70 | 11.45 | **18.74** | +7.29 | 18.26 | 12.63 | -5.63 | 16.01 | 9.11 | -6.90 |
| Group 4 | 10 | 77.41 | 77.43 | +0.02 | 72.49 | 72.45 | -0.04 | 75.84 | 75.97 | +0.13 |
| | 20 | 76.99 | 77.26 | +0.28 | 72.45 | 72.62 | +0.17 | 75.79 | 75.59 | -0.20 |
| | 30 | 76.68 | 77.01 | +0.33 | 72.24 | 72.40 | +0.16 | 75.62 | 75.58 | -0.04 |
| | 40 | 76.26 | 76.67 | +0.42 | 71.52 | **72.05** | +0.53 | 75.08 | 75.10 | +0.02 |
| | 50 | 75.82 | 75.88 | +0.06 | 70.56 | **71.46** | +0.90 | 74.24 | 74.52 | +0.28 |
| | 60 | 74.91 | 75.18 | +0.27 | 69.11 | **71.36** | +2.25 | 73.06 | **73.71** | +0.65 |
| | 70 | 73.18 | **73.97** | +0.79 | 67.41 | **69.96** | +2.55 | 71.25 | **72.32** | +1.07 |
| Group 5 | 10 | 77.03 | 77.45 | +0.43 | 72.02 | **72.84** | +0.82 | 75.17 | **76.15** | +0.98 |
| | 20 | 75.89 | **77.22** | +1.33 | 70.01 | **72.98** | +2.98 | 73.70 | **75.92** | +2.22 |
| | 30 | 74.60 | **76.51** | +1.91 | 67.00 | **72.41** | +5.41 | 71.67 | **75.15** | +3.48 |
| | 40 | 71.69 | **75.14** | +3.45 | 62.50 | **70.90** | +8.40 | 67.04 | **74.01** | +6.97 |
| | 50 | 66.46 | **71.03** | +4.58 | 58.16 | **66.45** | +8.29 | 61.54 | **69.32** | +7.78 |
| | 60 | 58.35 | **68.89** | +10.54 | 45.24 | **64.24** | +19.00 | 53.68 | **68.01** | +14.33 |
| | 70 | 44.50 | **61.99** | +17.48 | 34.30 | **55.93** | +21.64 | 41.11 | **61.33** | +20.22 |

Table 17: Data-free results on ResNet18 on CIFAR-100. Group 6-10.

| Group | Sparsity (%) | $\ell_1$(%) | IF ($\ell_1$)(%) | $\delta$(%) | Taylor(%) | IF (Taylor)(%) | $\delta$(%) | LAMP(%) | IF (LAMP)(%) | $\delta$(%) |
|---|---|---|---|---|---|---|---|---|---|---|
| Group 6 | 10 | 69.78 | **75.80** | +6.02 | 67.37 | **71.79** | +4.42 | 71.67 | **75.18** | +3.51 |
| | 20 | 55.99 | **72.57** | +16.57 | 63.01 | **71.59** | +8.57 | 66.87 | **73.75** | +6.88 |
| | 30 | 39.59 | **71.20** | +31.62 | 51.77 | **69.61** | +17.84 | 54.26 | **71.37** | +17.11 |
| | 40 | 29.47 | **64.98** | +35.51 | 34.22 | **65.32** | +31.10 | 41.69 | **67.75** | +26.06 |
| | 50 | 18.06 | **46.48** | +28.42 | 23.29 | **44.48** | +21.19 | 27.90 | **43.30** | +15.41 |
| | 60 | 3.61 | **42.87** | +39.26 | 16.55 | **34.34** | +17.78 | 14.76 | **46.28** | +31.52 |
| | 70 | 1.10 | **19.20** | +18.11 | 9.24 | 8.78 | -0.45 | 5.72 | **21.46** | +15.74 |
| Group 7 | 10 | 77.31 | 77.47 | +0.16 | 72.48 | 72.49 | +0.01 | 76.02 | 76.09 | +0.07 |
| | 20 | 76.84 | **77.38** | +0.54 | 72.34 | 72.47 | +0.13 | 75.23 | **76.21** | +0.98 |
| | 30 | 76.48 | **77.23** | +0.75 | 71.81 | 72.28 | +0.47 | 73.93 | **75.94** | +2.01 |
| | 40 | 75.50 | **77.12** | +1.61 | 71.21 | **72.19** | +0.98 | 71.25 | **75.92** | +4.67 |
| | 50 | 73.61 | **75.56** | +1.96 | 69.77 | **71.31** | +1.54 | 69.59 | **73.99** | +4.40 |
| | 60 | 71.09 | **76.02** | +4.92 | 67.41 | **71.05** | +3.65 | 65.50 | **74.40** | +8.90 |
| | 70 | 67.94 | **74.64** | +6.70 | 66.31 | **69.77** | +3.46 | 60.58 | **72.63** | +12.05 |
| Group 8 | 10 | 77.23 | 77.70 | +0.46 | 72.17 | 72.66 | +0.48 | 75.42 | **76.16** | +0.74 |
| | 20 | 76.92 | 77.23 | +0.31 | 70.49 | **72.16** | +1.67 | 72.48 | **75.58** | +3.11 |
| | 30 | 75.83 | 76.13 | +0.30 | 65.82 | **71.09** | +5.27 | 64.99 | **73.56** | +8.56 |
| | 40 | 73.12 | **73.97** | +0.85 | 59.01 | **68.63** | +9.62 | 58.02 | **69.22** | +11.20 |
| | 50 | **68.84** | 66.72 | -2.12 | 51.62 | **57.01** | +5.39 | **50.76** | 49.91 | -0.85 |
| | 60 | **59.29** | 57.90 | -1.38 | 41.72 | **51.94** | +10.22 | 34.89 | **36.93** | +2.04 |
| | 70 | **43.29** | 33.75 | -9.53 | 30.38 | **35.99** | +5.61 | 19.31 | **31.30** | +11.99 |
| Group 9 | 10 | 73.95 | **75.43** | +1.47 | 70.52 | **72.23** | +1.71 | 73.37 | **75.19** | +1.82 |
| | 20 | 69.46 | **72.05** | +2.59 | 68.42 | **71.35** | +2.93 | 65.49 | **72.39** | +6.90 |
| | 30 | 46.02 | **62.52** | +16.50 | 63.97 | **68.60** | +4.63 | 55.27 | **68.80** | +13.53 |
| | 40 | 30.89 | **55.22** | +24.33 | 52.59 | **63.11** | +10.52 | 33.60 | **63.24** | +29.64 |
| | 50 | 12.79 | **36.13** | +23.34 | **44.74** | 43.83 | -0.91 | 21.64 | **33.83** | +12.19 |
| | 60 | 4.10 | **29.84** | +25.73 | 28.89 | **47.17** | +18.29 | 8.03 | **20.18** | +12.15 |
| | 70 | 1.30 | **18.36** | +17.07 | 12.86 | **24.49** | +11.63 | 3.00 | **7.29** | +4.29 |
| Group 10 | 10 | 77.58 | 77.46 | -0.12 | 72.59 | 72.75 | +0.16 | 75.95 | 76.20 | +0.25 |
| | 20 | 77.23 | 77.47 | +0.24 | 72.57 | 72.33 | -0.24 | 75.64 | 75.98 | +0.34 |
| | 30 | 77.23 | 76.96 | -0.27 | **72.51** | 71.91 | -0.59 | 74.52 | **75.72** | +1.21 |
| | 40 | **76.99** | 75.82 | -1.17 | 72.23 | 72.25 | +0.02 | 71.33 | **74.07** | +2.74 |
| | 50 | **76.17** | 74.90 | -1.27 | **72.30** | 71.79 | -0.51 | 69.52 | 69.74 | +0.22 |
| | 60 | 75.71 | 75.51 | -0.20 | **71.94** | 70.98 | -0.96 | 66.93 | 67.14 | +0.21 |
| | 70 | **73.79** | 72.51 | -1.29 | **70.34** | 66.40 | -3.95 | 59.09 | **65.78** | +6.70 |

Table 18: Data-free results on ResNet50 on ImageNet. Group 0-9. Pruning criterion: CHIP (Sui et al., 2021).

| Group | Sparsity (%) | CHIP (%) | IF (CHIP)(%) | $\delta$(%) | Group | Sparsity (%) | CHIP (%) | IF (CHIP)(%) | $\delta$(%) |
|---|---|---|---|---|---|---|---|---|---|
| Group 0 | 10 | 74.74 | **75.51** | +0.77 | Group 5 | 10 | 74.52 | **75.44** | +0.92 |
| | 20 | 71.83 | **74.79** | +2.96 | | 20 | 72.98 | **74.81** | +1.83 |
| | 30 | 66.79 | **73.31** | +6.52 | | 30 | 71.31 | **74.07** | +2.76 |
| | 40 | 60.13 | **70.88** | +10.75 | | 40 | 69.39 | **72.64** | +3.25 |
| Group 1 | 10 | 74.67 | **75.36** | +0.69 | Group 6 | 10 | 74.27 | **75.38** | +1.11 |
| | 20 | 73.57 | **74.66** | +1.09 | | 20 | 72.39 | **74.68** | +2.29 |
| | 30 | 72.01 | **73.7** | +1.69 | | 30 | 69.78 | **73.73** | +3.95 |
| | 40 | 70.48 | **72.42** | +1.94 | | 40 | 67.51 | **72.41** | +4.9 |
| Group 2 | 10 | 74.67 | **75.29** | +0.62 | Group 7 | 10 | 67.98 | **73.11** | +5.13 |
| | 20 | 73.11 | **74.27** | +1.16 | | 20 | 58.45 | **70.21** | +11.76 |
| | 30 | 71.45 | **72.88** | +1.43 | | 30 | 42.37 | **65.51** | +23.14 |
| | 40 | 68.98 | **71.15** | +2.17 | | 40 | 26.56 | **60.62** | +34.06 |
| Group 3 | 10 | 74.95 | **75.5** | +0.55 | Group 8 | 10 | 75.39 | 75.62 | +0.23 |
| | 20 | 73.42 | **74.9** | +1.48 | | 20 | 74.83 | 75.32 | +0.49 |
| | 30 | 72.16 | **73.97** | +1.81 | | 30 | 74.59 | **75.12** | +0.53 |
| | 40 | 70.19 | **72.58** | +2.39 | | 40 | 73.41 | **74.71** | +1.3 |
| Group 4 | 10 | 74.65 | **75.24** | +0.59 | Group 9 | 10 | 75.43 | 75.67 | +0.24 |
| | 20 | 73.71 | **75.01** | +1.3 | | 20 | 74.9 | **75.41** | +0.51 |
| | 30 | 72.62 | **74.38** | +1.76 | | 30 | 74.55 | **75.32** | +0.77 |
| | 40 | 71.24 | **73.54** | +2.3 | | 40 | 74.08 | **75.12** | +1.04 |

Table 19: Data-free results on ResNet50 on ImageNet. Group 0-5.

| Group | Sparsity (%) | $\ell_1$(%) | IF ($\ell_1$)(%) | $\delta$(%) | Taylor(%) | IF (Taylor)(%) | $\delta$(%) | LAMP(%) | IF (LAMP)(%) | $\delta$(%) |
|---|---|---|---|---|---|---|---|---|---|---|
| Group 0 | 10 | 74.29 | **75.11** | +0.82 | 73.58 | **75.46** | +1.88 | 72.84 | **75.29** | +2.45 |
| | 20 | 71.74 | **74.35** | +2.61 | 68.36 | **74.30** | +5.95 | 68.11 | **73.84** | +5.73 |
| | 30 | 68.05 | **72.90** | +4.86 | 61.10 | **72.56** | +11.46 | 62.51 | **72.45** | +9.94 |
| | 40 | 61.71 | **70.66** | +8.95 | 46.87 | **69.73** | +22.86 | 54.38 | **70.23** | +15.85 |
| | 50 | 52.76 | **66.69** | +13.93 | 30.23 | **64.60** | +34.37 | 42.47 | **66.12** | +23.65 |
| | 60 | 41.90 | **62.18** | +20.28 | 17.64 | **59.17** | +41.53 | 29.79 | **59.71** | +29.93 |
| | 70 | 28.35 | **52.94** | +24.59 | 10.81 | **48.84** | +38.03 | 19.90 | **45.74** | +25.84 |
| Group 1 | 10 | 74.78 | **75.30** | +0.52 | 74.99 | 75.33 | +0.34 | 75.06 | 75.50 | +0.44 |
| | 20 | 73.82 | **74.60** | +0.77 | 73.92 | **74.77** | +0.85 | 74.14 | **74.75** | +0.61 |
| | 30 | 72.32 | **73.60** | +1.27 | 72.41 | **73.71** | +1.30 | 72.75 | **73.77** | +1.02 |
| | 40 | 70.93 | **72.33** | +1.41 | 71.01 | **72.60** | +1.59 | 70.78 | **72.69** | +1.91 |
| | 50 | 68.46 | **70.95** | +2.50 | 69.02 | **71.02** | +1.99 | 69.20 | **71.04** | +1.83 |
| | 60 | 65.88 | **68.46** | +2.58 | 65.97 | **68.29** | +2.32 | 66.37 | **68.86** | +2.50 |
| | 70 | 63.06 | **64.28** | +1.22 | 62.92 | **64.23** | +1.31 | 63.35 | **64.81** | +1.46 |
| Group 2 | 10 | 74.76 | **75.35** | +0.59 | 74.55 | **75.40** | +0.85 | 74.50 | **75.30** | +0.80 |
| | 20 | 73.61 | **74.60** | +0.99 | 73.08 | **74.45** | +1.37 | 73.08 | **74.45** | +1.37 |
| | 30 | 72.03 | **73.54** | +1.51 | 71.23 | **73.08** | +1.85 | 71.28 | **73.32** | +2.04 |
| | 40 | 70.13 | **71.97** | +1.84 | 69.19 | **71.37** | +2.18 | 68.91 | **71.80** | +2.89 |
| | 50 | 66.93 | **70.01** | +3.08 | 65.99 | **69.63** | +3.64 | 66.39 | **69.80** | +3.41 |
| | 60 | 63.41 | **66.70** | +3.29 | 62.15 | **65.67** | +3.52 | 63.36 | **66.14** | +2.77 |
| | 70 | 59.82 | **61.33** | +1.51 | 57.53 | **60.46** | +2.94 | 60.02 | **61.38** | +1.36 |
| Group 3 | 10 | 74.82 | **75.54** | +0.73 | 75.06 | **75.64** | +0.58 | 75.16 | 75.48 | +0.32 |
| | 20 | 73.84 | **75.04** | +1.20 | 74.34 | **75.01** | +0.67 | 74.37 | **75.16** | +0.79 |
| | 30 | 73.12 | **74.27** | +1.15 | 73.31 | **74.20** | +0.89 | 73.58 | **74.17** | +0.58 |
| | 40 | 72.23 | **73.18** | +0.95 | 72.21 | **73.41** | +1.20 | 72.47 | **72.99** | +0.52 |
| | 50 | 70.94 | 70.50 | -0.43 | 71.16 | 70.76 | -0.39 | **70.97** | 70.23 | -0.74 |
| | 60 | 69.53 | 69.09 | -0.44 | 69.76 | 69.57 | -0.19 | 69.01 | 68.87 | -0.14 |
| | 70 | **67.68** | 65.88 | -1.80 | **67.85** | 66.23 | -1.62 | **67.20** | 65.49 | -1.71 |
| Group 4 | 10 | 75.24 | 75.69 | +0.45 | 75.26 | 75.67 | +0.41 | 74.99 | **75.61** | +0.62 |
| | 20 | 74.54 | **75.32** | +0.79 | 74.28 | **75.24** | +0.97 | 73.99 | **75.15** | +1.16 |
| | 30 | 73.75 | **74.69** | +0.94 | 72.77 | **74.52** | +1.75 | 73.01 | **74.48** | +1.47 |
| | 40 | 72.75 | **73.88** | +1.13 | 70.47 | **73.73** | +3.26 | 71.95 | **73.53** | +1.58 |
| | 50 | 71.40 | 71.42 | +0.02 | 67.87 | **71.52** | +3.65 | 70.71 | 71.15 | +0.44 |
| | 60 | 69.64 | 69.91 | +0.27 | 64.26 | **69.72** | +5.46 | 68.38 | **69.54** | +1.16 |
| | 70 | **66.96** | 65.50 | -1.45 | 61.86 | **66.24** | +4.38 | 63.92 | 63.93 | +0.01 |
| Group 5 | 10 | 74.86 | **75.55** | +0.69 | 74.58 | **75.35** | +0.77 | 74.87 | **75.53** | +0.67 |
| | 20 | 73.61 | **75.02** | +1.41 | 73.08 | **74.93** | +1.85 | 73.67 | **74.97** | +1.29 |
| | 30 | 71.93 | **74.43** | +2.49 | 71.45 | **74.06** | +2.61 | 72.34 | **74.31** | +1.97 |
| | 40 | 70.25 | **73.22** | +2.97 | 69.40 | **72.87** | +3.47 | 70.06 | **73.06** | +3.00 |
| | 50 | 68.03 | **70.86** | +2.84 | 66.64 | **71.24** | +4.60 | 67.63 | **70.86** | +3.23 |
| | 60 | 64.61 | **68.41** | +3.80 | 63.67 | **68.82** | +5.15 | 64.74 | **68.07** | +3.33 |
| | 70 | 61.39 | **62.65** | +1.26 | 58.06 | **63.26** | +5.20 | 61.20 | **62.28** | +1.08 |

Table 20: Data-free results on ResNet50 on ImageNet. Group 6-11.

| Group | Sparsity (%) | $\ell_1$(%) | IF ($\ell_1$)(%) | $\delta$(%) | Taylor(%) | IF (Taylor)(%) | $\delta$(%) | LAMP(%) | IF (LAMP)(%) | $\delta$(%) |
|---|---|---|---|---|---|---|---|---|---|---|
| Group 6 | 10 | 74.56 | **75.43** | +0.87 | 74.37 | **75.30** | +0.93 | 74.79 | **75.46** | +0.67 |
| | 20 | 73.08 | **74.88** | +1.80 | 72.53 | **74.61** | +2.08 | 73.18 | **74.97** | +1.79 |
| | 30 | 70.98 | **74.09** | +3.11 | 70.66 | **74.02** | +3.36 | 71.28 | **74.09** | +2.82 |
| | 40 | 68.92 | **72.85** | +3.93 | 67.30 | **72.68** | +5.38 | 69.06 | **72.98** | +3.92 |
| | 50 | 66.28 | **70.65** | +4.37 | 62.90 | **70.82** | +7.91 | 66.23 | **70.53** | +4.29 |
| | 60 | 63.25 | **67.88** | +4.63 | 58.67 | **67.86** | +9.19 | 64.07 | **67.39** | +3.32 |
| | 70 | 59.09 | **61.80** | +2.71 | 52.85 | **60.48** | +7.63 | 58.50 | **61.42** | +2.91 |
| Group 7 | 10 | 71.18 | **74.31** | +3.13 | 72.77 | **74.68** | +1.91 | 72.90 | **75.15** | +2.25 |
| | 20 | 64.11 | **72.27** | +8.16 | 66.90 | **72.42** | +5.52 | 67.77 | **73.35** | +5.58 |
| | 30 | 50.33 | **69.02** | +18.69 | 57.71 | **68.86** | +11.14 | 59.52 | **70.93** | +11.41 |
| | 40 | 33.79 | **62.84** | +29.05 | 45.91 | **63.56** | +17.65 | 45.87 | **65.46** | +19.60 |
| | 50 | 18.54 | **51.01** | +32.47 | 33.06 | **51.04** | +17.98 | 29.57 | **46.38** | +16.81 |
| | 60 | 7.86 | **34.63** | +26.77 | 17.87 | **31.53** | +13.66 | 9.14 | **31.87** | +22.74 |
| | 70 | 2.49 | **7.10** | +4.61 | 6.70 | **7.40** | +0.70 | 2.21 | **6.02** | +3.81 |
| Group 8 | 10 | 75.51 | 75.80 | +0.29 | 75.63 | 75.79 | +0.17 | 75.61 | 75.78 | +0.17 |
| | 20 | 75.03 | 75.49 | +0.46 | 75.30 | 75.58 | +0.28 | 75.28 | 75.66 | +0.38 |
| | 30 | 74.43 | **75.24** | +0.80 | 74.75 | **75.28** | +0.54 | 74.64 | **75.22** | +0.58 |
| | 40 | 73.88 | **74.92** | +1.04 | 74.31 | 74.70 | +0.39 | 74.28 | 74.76 | +0.49 |
| | 50 | 72.86 | **73.47** | +0.60 | 73.47 | 73.37 | -0.10 | 73.46 | 73.41 | -0.05 |
| | 60 | 71.82 | **72.67** | +0.85 | 72.38 | 72.58 | +0.20 | **72.64** | 72.10 | -0.54 |
| | 70 | **70.69** | 70.06 | -0.63 | 70.79 | 70.58 | -0.21 | **71.13** | 68.54 | -2.59 |
| Group 9 | 10 | 75.74 | 75.85 | +0.11 | 75.58 | 75.72 | +0.14 | 75.72 | 75.82 | +0.10 |
| | 20 | 75.39 | 75.69 | +0.30 | 75.24 | 75.61 | +0.38 | 75.36 | 75.82 | +0.47 |
| | 30 | 75.00 | 75.41 | +0.41 | 74.90 | 75.36 | +0.46 | 74.82 | **75.48** | +0.66 |
| | 40 | 74.40 | **75.11** | +0.71 | 74.35 | **75.00** | +0.66 | 74.45 | **75.17** | +0.72 |
| | 50 | 73.95 | 74.02 | +0.07 | 73.87 | 74.14 | +0.27 | 73.79 | 74.11 | +0.32 |
| | 60 | 73.43 | 73.61 | +0.18 | 73.29 | 73.69 | +0.40 | 73.27 | **73.92** | +0.66 |
| | 70 | **72.86** | 71.83 | -1.02 | 72.54 | 72.33 | -0.21 | 72.38 | 71.95 | -0.43 |
| Group 10 | 10 | 75.56 | 75.72 | +0.16 | 75.80 | 75.75 | -0.04 | 75.69 | 75.84 | +0.16 |
| | 20 | 75.22 | 75.63 | +0.41 | 75.47 | 75.64 | +0.17 | 75.47 | 75.66 | +0.18 |
| | 30 | 75.01 | 75.39 | +0.38 | 75.17 | 75.46 | +0.29 | 75.23 | 75.49 | +0.26 |
| | 40 | 74.58 | 75.00 | +0.41 | 74.86 | 75.11 | +0.25 | 74.74 | 75.17 | +0.43 |
| | 50 | 74.27 | 74.46 | +0.19 | 74.40 | 74.13 | -0.27 | **74.61** | 73.96 | -0.65 |
| | 60 | 73.86 | 73.94 | +0.08 | 73.81 | 74.00 | +0.19 | **74.27** | 71.24 | -3.03 |
| | 70 | **73.42** | 72.44 | -0.97 | **73.42** | 72.45 | -0.96 | **73.59** | 68.29 | -5.30 |
| Group 11 | 10 | 75.75 | 75.88 | +0.14 | 75.70 | 75.77 | +0.07 | 75.73 | 75.91 | +0.18 |
| | 20 | 75.57 | 75.78 | +0.22 | 75.58 | 75.70 | +0.12 | 75.45 | 75.81 | +0.36 |
| | 30 | 75.25 | 75.70 | +0.45 | 75.41 | 75.50 | +0.09 | 75.10 | **75.66** | +0.56 |
| | 40 | 75.05 | 75.44 | +0.39 | 75.02 | 75.47 | +0.44 | 74.60 | **75.42** | +0.81 |
| | 50 | 74.62 | 74.66 | +0.04 | 74.79 | 74.76 | -0.03 | 74.13 | 74.59 | +0.47 |
| | 60 | 74.11 | 74.43 | +0.32 | 74.27 | 74.56 | +0.28 | 73.45 | **74.26** | +0.81 |
| | 70 | 73.32 | 73.19 | -0.14 | 73.66 | 73.54 | -0.12 | 72.80 | 72.91 | +0.12 |

Table 21: Data-free results on ResNet50 on ImageNet. Group 12-17.

| Group | Sparsity (%) | $\ell_1$(%) | IF ($\ell_1$)(%) | $\delta$(%) | Taylor(%) | IF (Taylor)(%) | $\delta$(%) | LAMP(%) | IF (LAMP)(%) | $\delta$(%) |
|---|---|---|---|---|---|---|---|---|---|---|
| Group 12 | 10 | 75.65 | 75.82 | +0.17 | 75.71 | 75.79 | +0.09 | 75.67 | 75.78 | +0.11 |
| | 20 | 75.41 | 75.71 | +0.30 | 75.36 | 75.69 | +0.32 | 75.40 | 75.59 | +0.19 |
| | 30 | 74.91 | **75.51** | +0.60 | 74.89 | **75.51** | +0.62 | 75.07 | 75.53 | +0.47 |
| | 40 | 74.44 | **75.14** | +0.70 | 74.50 | **75.26** | +0.76 | 74.45 | **75.05** | +0.60 |
| | 50 | 73.85 | 74.16 | +0.32 | 74.26 | 74.47 | +0.21 | 73.69 | 73.76 | +0.06 |
| | 60 | 72.92 | **73.63** | +0.71 | 73.38 | **74.07** | +0.69 | **72.75** | 71.93 | -0.82 |
| | 70 | **72.08** | 71.33 | -0.75 | 72.65 | 72.61 | -0.05 | **72.02** | 66.49 | -5.53 |
| Group 13 | 10 | 75.83 | 75.87 | +0.04 | 75.75 | 75.83 | +0.07 | 75.75 | 75.85 | +0.10 |
| | 20 | 75.62 | 75.78 | +0.15 | 75.62 | 75.74 | +0.12 | 75.59 | 75.81 | +0.22 |
| | 30 | 75.43 | 75.77 | +0.34 | 75.53 | 75.81 | +0.29 | 75.21 | **75.73** | +0.51 |
| | 40 | 74.90 | **75.51** | +0.61 | 75.11 | 75.54 | +0.43 | 74.86 | **75.58** | +0.72 |
| | 50 | 74.59 | 74.77 | +0.18 | 74.87 | 74.79 | -0.08 | 74.30 | 74.55 | +0.24 |
| | 60 | 73.97 | 74.25 | +0.28 | 74.25 | 74.41 | +0.16 | 73.67 | **74.19** | +0.52 |
| | 70 | **73.07** | 72.01 | -1.07 | **73.36** | 72.66 | -0.70 | **72.80** | 72.22 | -0.59 |
| Group 14 | 10 | 75.72 | 75.80 | +0.08 | 75.80 | 75.76 | -0.04 | 75.53 | 75.77 | +0.24 |
| | 20 | 75.40 | 75.65 | +0.25 | 75.56 | 75.73 | +0.17 | 75.32 | 75.67 | +0.35 |
| | 30 | 75.06 | 75.51 | +0.45 | 75.34 | 75.64 | +0.30 | 75.06 | 75.49 | +0.44 |
| | 40 | 74.45 | **75.15** | +0.70 | 74.97 | 75.24 | +0.27 | 74.69 | **75.22** | +0.53 |
| | 50 | 73.63 | 74.13 | +0.50 | 74.67 | 74.42 | -0.25 | 74.05 | 74.18 | +0.12 |
| | 60 | 72.92 | 73.29 | +0.38 | 73.90 | 73.93 | +0.03 | 73.11 | 73.46 | +0.34 |
| | 70 | **72.28** | 70.39 | -1.89 | 72.95 | 72.50 | -0.45 | **72.12** | 69.76 | -2.36 |
| Group 15 | 10 | 75.84 | 75.94 | +0.10 | 75.68 | 75.89 | +0.21 | 75.69 | 75.81 | +0.13 |
| | 20 | 75.59 | 75.87 | +0.28 | 75.60 | 75.77 | +0.17 | 75.55 | 75.76 | +0.22 |
| | 30 | 75.16 | **75.67** | +0.52 | 75.41 | 75.70 | +0.29 | 75.27 | 75.75 | +0.48 |
| | 40 | 74.58 | **75.41** | +0.83 | 75.15 | 75.43 | +0.29 | 74.79 | **75.42** | +0.63 |
| | 50 | 73.87 | 74.37 | +0.50 | 74.80 | 74.30 | -0.50 | 74.13 | 74.37 | +0.25 |
| | 60 | 72.90 | **73.49** | +0.60 | **74.30** | 73.71 | -0.59 | 73.45 | 73.82 | +0.37 |
| | 70 | **71.81** | 70.31 | -1.50 | **73.70** | 71.03 | -2.67 | **71.91** | 70.89 | -1.02 |
| Group 16 | 10 | 75.71 | 75.83 | +0.12 | 75.75 | 75.90 | +0.15 | 75.63 | 75.87 | +0.24 |
| | 20 | 75.39 | 75.68 | +0.29 | 75.51 | 75.79 | +0.29 | 75.48 | 75.83 | +0.35 |
| | 30 | 75.07 | 75.49 | +0.43 | 75.12 | 75.60 | +0.48 | 75.08 | 75.54 | +0.46 |
| | 40 | 74.37 | **75.18** | +0.81 | 74.21 | **75.15** | +0.94 | 74.74 | 75.16 | +0.42 |
| | 50 | 73.04 | **73.99** | +0.95 | 72.51 | **73.85** | +1.34 | 74.01 | 74.04 | +0.03 |
| | 60 | 70.39 | **73.31** | +2.92 | 68.92 | **73.52** | +4.59 | 73.51 | 73.03 | -0.47 |
| | 70 | 67.94 | **70.54** | +2.60 | 64.13 | **70.42** | +6.29 | **72.07** | 69.05 | -3.02 |
| Group 17 | 10 | 75.59 | 75.84 | +0.25 | 75.74 | 75.79 | +0.04 | 75.83 | 75.94 | +0.12 |
| | 20 | 75.31 | **75.84** | +0.53 | 75.57 | 75.72 | +0.15 | 75.55 | 75.86 | +0.31 |
| | 30 | 74.68 | **75.61** | +0.93 | 75.28 | 75.72 | +0.44 | 74.90 | **75.67** | +0.77 |
| | 40 | 73.20 | **75.25** | +2.05 | 74.81 | **75.40** | +0.59 | 73.59 | **75.07** | +1.48 |
| | 50 | 70.32 | **73.60** | +3.28 | 74.25 | 74.08 | -0.17 | 70.47 | **73.10** | +2.63 |
| | 60 | 65.56 | **72.76** | +7.20 | **73.45** | 72.87 | -0.58 | 65.95 | **71.47** | +5.52 |
| | 70 | 56.26 | **67.58** | +11.33 | **71.67** | 68.94 | -2.73 | 59.27 | **62.87** | +3.60 |

