# OpenReview forum: "Towards Meta-Pruning via Optimal Transport"
_ICLR.cc/2024/Conference — ICLR 2024 spotlight_

### Official Review · Reviewer_ZhLo · 2023-10-30

**Soundness:** 3 good
**Presentation:** 3 good
**Contribution:** 3 good
**Rating:** 8
**Confidence:** 3

**Summary:**

This paper proposes a new intra-fusion approach, which tries to unify and bridge the paradigms of pruning and fusion. The proposed approach shows considerable gains with or without fine-tuning.

**Strengths:**

* The idea of unifying pruning and fusion into a principled approach is very interesting, and makes a lot of sense.
* The argument that we should not just keep the most important nodes while discarding the others, but can actually restore information from all neurons to create more accurate compressed networks is novel.
* The evaluation is comprehensive and interesting, while the analysis (Sec 5) demonstrates many interesting findings.
* The discussion on applications beyond pruning, i.e., factorizing model training seems to be viable.

**Weaknesses:**

* It is not clear to me why uniform distribution for both source and target is generally the most robust choice? (Sec 3.2.3)
* Is that possible to use existing pruning-at-initialization metrics, such as SynFlow or ZiCo, with the proposed Intra-Fusion?
* In the split-data approach, it is mentioned that if we train two individual models, the speed-up would (in theory) be 2x. Can you demonstrate this with some experimental results in practice?

**Questions:**

See above.

---

> ### Author Response · Authors · 2023-11-20
>
> Thank you for taking the time to review our paper and for providing valuable feedback. We are pleased to hear that you find the idea of unifying pruning and fusion into a principled approach interesting and sensible. Your positive comments on the novelty of restoring information from all neurons to create more accurate compressed networks are greatly appreciated. We also appreciate your acknowledgment of the comprehensive and interesting evaluation, as well as the noteworthy findings presented in Section 5.
>
> We would like to address the specific points you raised as weaknesses:
>
> ---
>
> **Robustness of uniform distributions:**
>
> The choice of uniform probabilities is rooted in considerations of entropy, which as considered in information theory, is a measure of uncertainty or disorder in a system. In the context of Optimal Transport, using uniform probabilities for both source and target distributions maximizes entropy and, consequently, represents a **conservative** approach that minimizes assumptions about the underlying structures of the distributions amidst uncertainty about true but unknown importance distributions.
>
> In hindsight, we do agree that the formulation in the paper is a bit vague, and have thus reformulated it in the revised version (see Section 3.2.3, page 5). Furthermore, to add some further exploration with regards to the varying performance for different choices of target and source distributions, we added a corresponding ablation study in Appendix C.4. Thank you for bringing this to our attention!
>
> ---
>
> **Intra-Fusion & Pruning-at-Initialization metrics:**
>
> It would indeed be fascinating to explore the integration of Intra-Fusion within a data-free pruning-at-initialization (PAI) framework, such as with Synflow. Regrettably, PAI has predominantly focused on unstructured pruning rather than the structured pruning paradigm that Intra-Fusion addresses. For example, Synflow is designed for unstructured pruning, and the problem it intends to alleviate (layer collapse) does not have a direct parallel in structured pruning.
>
> Nevertheless, as pointed out before, it would indeed be very interesting to combine these two lines of work. Most PAI approaches derive “winning tickets” by relying on the assumption that weights/neurons are discrete and separate entities, whereas Intra-Fusion posits that akin neurons can be merged. This could introduce a novel dimension to PAI wherein one could try to fuse the lost information from the dense ticket back into the chosen sparse mask — something which, to the best of our knowledge, remains unexplored and would form an interesting direction for future work.
>
> ---
>
> ## Experimental Results on Speedups of the Split-Data Approach
>
> **Additional figure with timeline comparison:**
>
> We want to thank the reviewer for bringing to our attention that adding a concrete run-time comparison of split-data approaches and the whole data approach is of interest and will again clarify the chosen PaF/FaP methodology. Hence, we have added an additional figure in the Appendix D.1 giving a concrete side-by-side comparison of the timelines of the approaches and providing concrete speedup figures.
>
> **Practical speedup measures on VGG11-BN (training:1.81, PaF:1.42, FaP:1.31):**
>
> In the added figure we observe a practical speedup for a VGG11-BN (CIFAR10) of the original model training time of 1.81. Here the overall speedups of the split-data approaches are 1.42 (PaF), 1.31 (FaP). The increase in speedup of PaF over FaP is due to the fine-tuning being done on an already pruned network, thus making use of the improved inference time that comes with structurally pruned networks.
>
> **Training factorization yields higher speedups when train time >> fine-tune time:**
> It is important to note that the split-data approaches presented are proof of the concept that model training can be effectively factorized using model fusion and pruning. By optimizing hyperparameters like “number of fine-tuning epochs” at different stages of the approaches further speedups are possible. For models and datasets where the time for original training $T_1$ is much greater than time for fine-tuning $T_2$ $(T_1 >> T_2)$, speedups would show to be much more significant.
>
> ---------------------
> Thank you once more for investing your time and effort in reviewing our paper! We hope that we could provide further clarity on the choice of source and target distributions, and the distinctions between Intra-Fusion and PAI.

---

### Official Review · Reviewer_vYA3 · 2023-10-31

**Soundness:** 3 good
**Presentation:** 3 good
**Contribution:** 3 good
**Rating:** 8
**Confidence:** 5

**Summary:**

I find this paper to be intriguing and appreciate its unique approach, diverging from traditional methods by employing optimal transport. The performance of this method appears promising, and I commend the authors for their work. I am inclined to give this paper a high rating and would recommend its acceptance.

**Strengths:**

I would like to express my sincere appreciation for the data-free setup presented in this paper; it is an aspect that I find to be incredibly valuable. The capability of the proposed method to effectively operate within a data-free setup is both intriguing and commendable.



I am genuinely intrigued by the concept of split-data training introduced in the paper. Its novelty, practicality, and engaging nature hold great potential to inspire and contribute significantly to the broader community.


The proposed method skillfully utilizes a modified version of OTFusion to integrate the discarded neurons into the "surviving" ones, a strategy I find to be quite thoughtful and well-reasoned.


A significant portion of the research in structured pruning has concentrated on creating more significant importance measures, denoted as i, while the overarching procedure outlined in Algorithm 1 has largely stayed consistent. Drawing inspiration from OTFusion, this paper endeavors to explore an innovative approach to the conventional pruning method. Rather than merely eliminating the less crucial pairings within a group, the authors thoughtfully utilize the calculated importance metrics to guide the fusion of these pairings, ultimately resulting in a reduced group cardinality. This nuanced method demonstrates a commendable attempt to enhance the efficiency of structured pruning.

The results showcased by the proposed method, particularly within the data-free setup, are indeed promising and show great potential. This outcome is both encouraging and exciting, and it highlights the method's effectiveness in challenging scenarios.


The results presented in Figure 6 appear to be promising and demonstrate the potential effectiveness of the approach being discussed.

**Weaknesses:**

I kindly draw your attention to pages 3 and 4, where multiple references to “Figure 6” appear. It is possible that these may be typographical errors, and you might be intending to refer to “Figure 1” instead.

It would be beneficial if the authors could extend their comparisons in the main text to include a wider range of prior works, in addition to the baseline they have already examined. This would provide a more comprehensive understanding of how the proposed method stands in relation to existing literature, and it would undoubtedly enrich the paper's overall context and value.

**Questions:**

See #Weakness.

---

> ### Author Response · Authors · 2023-11-20
>
> We would like to express our gratitude for your thoughtful and constructive review of our paper. Your positive feedback regarding the data-free setup and the novelty of the split-data training approach is truly encouraging, and we appreciate your acknowledgment of the potential impact our method may have on the broader research community.
>
> Your recognition of our strategy involving the integration of discarded neurons using a modified version of OTFusion is particularly satisfying. We aimed to provide a well-reasoned and thoughtful approach to structured pruning, and we are pleased to see that our efforts in this regard have resonated with you.
>
> ---
>
> **Extension of prior work:**
>
> We also appreciate your insightful comments on the comparison of our method with existing literature. Your suggestion to extend our comparisons to include a wider range of prior works is well-taken. To address it, we added experimental results of deploying the importance metric in [1] (see Appendix E.2, Table 18). To further highlight that the improvements of Intra-Fusion are agnostic to the choice of importance metric, we also include results of using random scores drawn from a uniform distribution as an importance metric (see Appendix C.3).
>
> ---
>
> **Writing enhancements:**
>
> Regarding your note on the typographical errors in references to Figure 6 on pages 3 and 4, we sincerely apologize for any confusion caused by this oversight. We have since corrected these errors and ensured that the references accurately point to Figure 1.
>
> ---------------------
>
> Once again, we appreciate the time and effort you have dedicated to reviewing our paper, and we hope to have addressed your suggestions adequately.
>
> [1] CHIP: CHannel independence-based pruning for compact neural networks. NeurIPS, 2021, Yang Sui, Miao Yin, Yi Xie, Huy Phan, Saman Aliari Zonouz, Bo Yuan.

---

### Official Review · Reviewer_u8hy · 2023-11-01

**Soundness:** 3 good
**Presentation:** 3 good
**Contribution:** 3 good
**Rating:** 6
**Confidence:** 4

**Summary:**

The paper introduces 'Intra-Fusion', a novel approach that combines the concepts of pruning and fusion for compressing over-parameterized neural networks. While pruning is a well-established method for reducing the size of neural networks, fusion, which involves merging independently trained networks, has recently gained traction. The authors propose a method that leverages pruning criteria to inform the fusion process. This approach, irrespective of the specific neuron-importance metric used, can prune a significant number of parameters while maintaining accuracy levels comparable to standard pruning methods. Furthermore, the paper explores how fusion can enhance the pruning process, reducing training time without compromising performance. The results are benchmarked on popular datasets like CIFAR10, CIFAR100, and ImageNet.

**Strengths:**

1. The paper is well-organized and clearly written, which is easy to follow.
2. The problem studied in this paper is interesting and valuable.
3. The experimental verification is quite sufficient.

**Weaknesses:**

1. This paper presents extensive experiments across various settings. However, there are areas that could benefit from further exploration: It would be valuable to see comparative results with methods like the LOTTERY TICKET HYPOTHESIS [A]. How does the proposed approach stack up against such established techniques?
2. The FaP approach introduced in this paper seems to assume that the two models being fused have identical structures. It raises the question of how adaptable this method is. Can it handle fusions of models with different depths or even heterogeneous models, such as combining a CNN with a ViT[B]?

[A] Jonathan Frankle and Michael Carbin. The lottery ticket hypothesis: Finding sparse, trainable neural
networks. ICLR 2019.
[B] Alexey Dosovitskiy, Lucas Beyer, Alexander Kolesnikov, Dirk Weissenborn, Xiaohua Zhai, Thomas Unterthiner, Mostafa Dehghani, Matthias Minderer, Georg Heigold, Sylvain Gelly, Jakob Uszkoreit, Neil Houlsby. An Image is Worth 16x16 Words: Transformers for Image Recognition at Scale. ICLR2021.

**Questions:**

Please see the Weaknesses.

---

> ### Author Response · Authors · 2023-11-20
>
> Thank you for the thorough reading and the feedback on our work.
>
> ---
>
> **Comparison to performance of Lottery Ticket Hypothesis (LTH):**
>
> We would like to highlight that LTH and Intra-Fusion address distinct aspects of neural network optimization.
>
> Our work deals with structured pruning, which involves removing entire neurons based on an importance metric to compress already trained large neural networks. On the other hand, the Lottery Ticket Hypothesis posits that there exist sparse **unstructured** subnetworks within larger neural networks that, when trained in isolation, can achieve comparable performance. Moreover, LTH is strictly speaking **not a deployable pruning methodology**, for it offers a ‘retrospective’ method for which the entire network training has to be, repeatedly, carried from scratch until the pruning mask of desired sparsity is obtained. This is because,
> * it requires the availability of the network checkpoint at an earlier epoch, if not the initialization, to apply the intermediate pruning masks. This is rarely the case in practice.
> * Besides, it also assumes the original training dataset (or some additional dataset) is available *(not data-free)*. Compared to Intra-Fusion, this adds to computational (due to repeated training) and data procurement costs.
>
> As evident from the above discussion,  LTH is concerned with unstructured pruning, barring odd exceptions, as opposed to the structured pruning paradigm which Intra-Fusion addresses. A focus on structured pruning lets us deliver immediate speedup in inference and storage costs, while unstructured pruning is not directly amenable to hardware acceleration.
>
> Nevertheless, we do agree that our paper can only be enriched by an even broader comparison. For this reason, in the new version of the paper, we also compare Intra-Fusion with an additional recent top-performing importance metric CHIP in [1] (see Appendix E.2, Table 18). Moreover, to highlight the point that Intra-Fusion is agnostic to the importance criterion, we include results with random pruning (see Appendix C.3).
>
> ---
>
> **Adaptivity of the FaP approach:**
>
> Thank you for bringing up this interesting point. Research on model fusion is still in a relatively nascent stage, and fusion techniques such as Vanilla Averaging (which is a key ingredient of Federated Averaging) require identical model architectures. However, the model fusion technique we use in our split-data method, namely OTFusion [2], is able to fuse models with different widths. Hence, fusing model architectures that exhibit some degree of heterogeneity is still possible.
>
> However, fusing models of radically heterogeneous architectures, such as CNNs with ViTs, as you point out, is still an unexplored area, and conceptually very complex. However, one could potentially identify similar layers and fuse them within two models of highly heterogeneous architectures, thereby yielding a child network that is separate in some layers, and conjoined in others.
>
> ---------------------
> In conclusion, we appreciate the valuable feedback from the reviewer, which has enhanced the clarity of our paper. We've clarified the distinctions between Intra-Fusion and the Lottery Ticket Hypothesis, and provided a broader comparison in the revised version (Table 18, and Appendix C.3).
>
> Regarding the adaptivity of our FaP approach, we acknowledge the complexity of fusing models with radically different architectures. We've discussed the challenges and potential directions for future exploration.
> We kindly request the reviewer to reconsider their evaluation in light of the addressed concerns and the enriched comparison.
>
> [1] CHIP: CHannel independence-based pruning for compact neural networks. NeurIPS, 2021, Yang Sui, Miao Yin, Yi Xie, Huy Phan, Saman Aliari Zonouz, Bo Yuan.
>
> [2] Model Fusion via Optimal Transport . NeurIPS, 2020, Sidak Pal Singh, Martin Jaggi.

---

### Official Review · Reviewer_t7SM · 2023-11-02

**Soundness:** 3 good
**Presentation:** 3 good
**Contribution:** 3 good
**Rating:** 8
**Confidence:** 5

**Summary:**

This paper presents a new model compression technique, IntraFusion, by combining pruning and merging (or fusion). The fusion part is based on Optimal Transport (OT). The main idea is to combine multiple independently trained neural networks. Empirical results suggest that IntraFusion performs better than the default pruning scheme, especially in the no finetuning ("data-free pruning") setting.

**Strengths:**

1. Most network pruning methods still rely on an excessive retraining process. This paper proposes a method to save the retraining, which potentially is of broad interest.

2. The proposed method uses OT to merge networks for model compression, unlike most of the conventional ways, which sounds novel to me.

3. The empirical results suggest the method is more effective than the default pruning scheme, especially without finetuning.

**Weaknesses:**

1. My biggest concern is about the empirical results.

1.1 Currently, it only compares with the default pruning for the main benchmark results (Tab. 1, Fig. 3 and 4). This looks quite limited to me. How is the method compared to other recent top-performing structured pruning methods like [*1 - *3]? It is highly advisable to add a set of comparisons with ResNet50 on ImageNet (as far as I know, this is the standard benchmark setup in a typical pruning paper).

1.2 Based on Tab. 1, after finetuning, the advantage of the proposed method seems quite marginal (the authors also agree that "*the gains might not seem as stark*"). Although the authors argue that "the focus of this paper is on data-free pruning", I do not think this is strong enough to justify the weak improvement, since  "data-free pruning" is barely practical at present. This means, when used in practice, the proposed method actually will not see much advantage against the default pruning scheme.

2. Some minor issues:
* "Pruning techniques (LeCun et al., 1989) can broadly be classified into structured (Wang et al., 2019; Frantar & Alistarh, 2023)," -- The paper (Frantar & Alistarh, 2023) seems not to be structured pruning paper. It is unstructured.

References:

- [*1] Neural pruning via growing regularization. ICLR, 2021.
- [*2] CHIP: CHannel independence-based pruning for compact neural networks. NeurIPS, 2021.
- [*3] Trainability preserving neural pruning. ICLR, 2023.

=== Post Rebuttal ===
The authors have well-addressed my concerns. The new results with CHIP and random pruning criterion show the efficacy of the method on ResNet50 with ImageNet, rating thus raised from 6 to 8.

**Questions:**

What is the training cost of the method compared to the default pruning scheme?

---

> ### Author Response · Authors · 2023-11-20
>
> We thank the reviewer for the detailed feedback and for recognizing the benefits of combining model pruning and model fusion.
>
> ---
>
> > How is the method compared to other recent top-performing structured pruning methods
>
> **Intra-Fusion challenges the default meta-pruning approach:**
>
> It is important to note that we are not competing with any specific pruning methodology per se, but with the overarching “meta-pruning” approach, i.e. how a given importance metric is used.
>
> **Validation of Intra-Fusion on CHIP and random importance metric:**
>
> However, we are happy to further validate that Intra-Fusion manages to leverage a given importance metric more effectively than the traditional approach of simply discarding the least important neuron pairings. Hence, we added experimental results of deploying the importance metric in [1] (see Appendix E.2, Table 18). To further highlight that the improvements of Intra-Fusion are agnostic to the choice of importance metric, we include results of choosing importance at random (see Appendix C.3).
>
> ---
>
> > The advantage of the proposed method seems quite marginal
>
> The significance of the proposed data-free pruning method lies in its ability to alleviate the resource-intensive nature of traditional model pruning. Nowadays pre-training large models on extensive data can be extremely laborious and confined to a selected few that have access to extensive resources. In many cases, the models are open-sourced, but the data used for pre-training are not. Providing a data-free way to compact these models without the need for extensive re-training can have large implications, particularly where data privacy concerns, resource constraints, or domain-specific limitations may hinder the availability of training data.
>
> ---
>
> >The paper (Frantar & Alistarh, 2023) [2] seems not to be structured pruning paper. It is unstructured.
>
> Thank you for bringing this to our attention! In the revised version we have replaced the citation with [3], which unequivocally concerns structured pruning.
>
> ---
>
> > Question: Training cost of the method compared to the default pruning scheme
>
> **Algorithmic complexity as in OTFusion:**
>
> Regarding a concrete analysis of our Intra-Fusion approach, we would like to kindly refer to the analysis done in the original OTFusion paper [4] as our Intra-Fusion approach exhibits the same algorithmic complexity.
>
> **Temporal cost of Intra-Fusion seems negligible in face of training times:**
>
> In case you are interested in concrete runtime figures from practice, we would like to also provide these. Deploying pruning via Intra-Fusion (without iterative retraining) in our pipeline takes about 3.78s. The default pruning scheme in contrast runs in about 2.45s. In the context of training, fine-tuning and iterative pruning (with fine-tuning) we find this increased temporal cost to be negligible.
>
> ---------------------
>
> In conclusion, we sincerely appreciate the reviewer's insightful feedback and thoughtful recognition of the strengths of our work. We have taken great care to address the concerns raised, providing additional experimental results and clarifications. We kindly request the reviewer to reconsider their evaluation in light of these improvements.
>
> [1] CHIP: CHannel independence-based pruning for compact neural networks. NeurIPS, 2021, Yang Sui, Miao Yin, Yi Xie, Huy Phan, Saman Aliari Zonouz, Bo Yuan.
>
> [2] SparseGPT: Massive Language Models Can Be Accurately Pruned in One-Shot. ArXiv, 2023, Elias Frantar, Dan Alistarh.
>
> [3] Towards Any Structural Pruning. CVPR, 2023, Gongfan Fang, Xinyin Ma, Mingli Song, Michael Bi Mi, Xinchao Wang.
>
> [4] Model Fusion via Optimal Transport . NeurIPS, 2020, Sidak Pal Singh, Martin Jaggi.

---

### Official Review · Reviewer_hkmL · 2023-11-07

**Soundness:** 2 fair
**Presentation:** 1 poor
**Contribution:** 2 fair
**Rating:** 6
**Confidence:** 1

**Summary:**

This paper focuses on the integration of pruning and fusion techniques in model compression. The authors introduce a novel approach called Intra-Fusion, which leverages Optimal Transport to inform the model compression process. Intra-Fusion aims to preserve the output of the original non-pruned model and recover accuracy without the need for fine-tuning or data. The paper also explores the application of fusion in factorizing and speeding up the training process of models that are pruned after training. Experimental results show that Intra-Fusion achieves consistent gains in test accuracy compared to conventional pruning methods.

**Strengths:**

1. Improved accuracy: The article demonstrates that Intra-Fusion can significantly enhance the accuracy of pruned models without relying on any additional data. By merging similar neurons, Intra-Fusion better preserves the output of the original non-pruned model, leading to superior performance.

2. Data-free pruning: Pruning neural networks usually results in immediate drops in accuracy, requiring extensive fine-tuning. However, the article argues that with Intra-Fusion, a significant amount of accuracy can be recovered without the need for any data points. This approach provides a more efficient and practical solution for model compression.

3. Factorizing model training: The article explores how fusion can be used to factorize and speed up the training process of models that are supposed to be pruned after training. By splitting the training dataset into subsets and training models concurrently, significant training time speedups can be achieved. This approach provides an alternative or enhancement to data parallelism during distributed model training.

**Weaknesses:**

1. This article has severe writing issues:
> + In the second paragraph of section 3.1, Figure 6 appears multiple times. I believe it should be Figure 1.
> + In Algorithm 1, $ neuron\ j \in layer ∧ i\[j\] \ge t$ represents a logical "AND" relationship, not a neuron.
> + The text contains many long and heavily clause-laden sentences, which pose a significant obstacle to understanding the article. I suggest avoiding such expressions as much as possible in academic papers, for example, in the sentence on the third line of Section 6.
> + The text description of Figure 7 has been truncated.

**Questions:**

1. What does "meta" manifest in?
2. In the fourth paragraph of the "Meta-Pruning Comparison" part in Section 3.1, it mentions "layer's cardinality." However, in the "Structured Pruning: Group-by-Group" part, it states, "The number of neuron pairings in a group we term 'group cardinality'." So, is 'cardinality' defined in the context of layers or groups?

---

> ### Author Response · Authors · 2023-11-20
>
> We thank the reviewer for the detailed feedback and for recognizing the benefits of combining model pruning and model fusion.
>
> ---
>
> **Writing Improvements:**
>
> Specifically, we would like to thank the reviewer for directing our attention towards the listed writing issues. We have gladly taken this as an opportunity to improve the quality and readability of our paper. In particular, to further the clarity of our approach, we have corrected the mentioned issues and rewritten the cause-laden sentences that we could identify. Please inform us if there are any remaining sections that are still difficult to understand so that we can remove any possible ambiguity.
>
> ---
>
> >What does "meta" manifest in?
>
> **The “meta-level” of deploying importance metrics:**
>
> By  “meta” in the context of “meta-pruning”, we refer to the process of taking a step back and considering what happens on a “higher/meta level” when doing pruning. Specifically, in the context of pruning the current paradigm is: keep the most important neuron pairings in a group and discard the rest. While most research has gone into developing more sophisticated and effective importance metrics, the overlying “meta-level” of leveraging these importance metrics has stayed the same. Intra-Fusion tries to challenge this and proposes an alternative way of utilizing an agnostically given importance metric.
>
>
> **Intra-Fusion leverages an agnostic importance metric more effectively:**
>
> Namely, we propose a new “meta-pruning approach” that instead of discarding the less important neuron pairings, integrates their learned features into the more important neuron pairings. To further highlight that Intra-Fusion is a more effective way of leveraging any agnostically given importance metric, we additionally include experimental results of using random scores drawn from a uniform distribution as an importance metric (see Section C.3).
>
> ---
>
> >So, is 'cardinality' defined in the context of layers or groups?
>
> We want to thank the reviewer for this question. The word “(group) cardinality” stands for the number of neuron pairings in a group, where all neurons contained in a neuron pairing have to be handled in unison. We realize that mentioning the concept of cardinality in the context of layers and in the context of groups might be confusing. In the updated version we now solely refer to “group cardinality”.
>
> ---------------------
>
> Please, let us know in case there are any further questions or comments. We will be more than happy to answer them or add further clarifications. Otherwise, in light of the above, we hope you will give your evaluation of the paper a second thought. Thank you once again for taking the time and we remain at your disposal.

---

> > ### Comment · Reviewer_hkmL · 2023-11-23
> > **Feedback to the author‘s response**
> >
> > Thanks for the reply. I hope that in future releases, efforts will be made to enhance the overall writing quality. The caliber of writing significantly influences the initial impression and comprehension of an article. A well-crafted academic essay demands not only a strong idea but also a high standard of writing. I will change my score.

---

### Author Response · Authors · 2023-11-20

We would like to thank all the reviewers for taking the time to review our work and for providing valuable feedback. Here, we want to address common points raised and give insights on further experiments that we have added in the context of the rebuttal — further strengthening our findings.

---

**More comparisons with SOTA pruning techniques:**

To further validate the advantageous utility of Intra-Fusion in leveraging an agnostically provided importance metric compared to the current paradigm, we have incorporated results for CHIP [1] in Appendix E.2, Table 18 (ResNet50 on ImageNet). Our findings demonstrate substantial gains in the data-free context, which aligns well with the outcomes observed for other importance metrics.

**Agnosticism to importance metrics:**

As elucidated in the paper and reiterated in this rebuttal, we argue that Intra-Fusion is agnostic regarding the choice of the importance metric. Emphasizing this stance, we present evidence in Appendix C.3 showcasing Intra-Fusion's efficacy to achieve significant accuracy gains — even when using random scores drawn from a uniform distribution as an importance metric. Essentially, Intra-Fusion helps even when lacking meaningful information about the importance of neural pairings, by inherently trying to preserve the function output. Therefore, the gain in accuracy provided by it fills in for less expressive importance metrics.

**Ablation study on varying source and target distribution in the OT setting:**

To better understand the impact of source and target distribution choices in the Optimal Transport (OT) setting, we conduct an ablation study in Appendix C.4. The results indicate that the choice of distribution does not yield statistically significant differences. In cases of improvements, they are only marginal.

**Concrete Runtime Comparison of Split-Data and Whole-Data:**

We have included an additional figure in the Appendix D.1 giving a concrete side-by-side comparison of the timelines of the approaches. On VGG11-BN (CIFAR10) we observe a practical speedup of the original model training time of $1.81\times$ (compared to the previously mentioned theoretical speedup of $2\times$). Here, the overall speedups of the split-data approaches are $1.42\times$ (PaF), $1.31\times$ (FaP). For models and datasets where the time for original training $T_1$ is much greater than the time for fine-tuning $T_2$ $(T1 >> T2)$, speedups are much more significant.

**Writing enhancements:**

We've carefully reviewed and refined the manuscript to uphold its overall quality standards. We trust that these enhancements contribute to a heightened clarity in our paper.

---------------------

[1] CHIP: CHannel independence-based pruning for compact neural networks. NeurIPS, 2021, Yang Sui, Miao Yin, Yi Xie, Huy Phan, Saman Aliari Zonouz, Bo Yuan.

---

### Meta-Review · Area_Chair_bsv1 · 2023-12-08

**Metareview:**

The authors propose a novel approach for model compression, namely IntraFusion, by leveraging optimal transport (OT) to combine pruning and merging. The proposed approach does not require fine tuning, but still maintain high performance. The Reviewers agree that this is a good submission for ICLR. The proposed approach is novel, clever to pruning to guide the merging in IntraFusion. The authors thoroughly show advantages of proposed approach.

**Justification For Why Not Higher Score:**

+ The authors propose a novel approach, IntraFusion for model compression. The authors cleverly leverage pruning to guide the merging. However, the idea of its merging is still heavily built upon the OT-Fusion in the literature. I think it is a good submission, but I feel that it may not be enough for oral acceptance yet.

**Justification For Why Not Lower Score:**

+ The reviewers agree that it is a good submission. The authors proposed interesting ideas to combine pruning and merging for model compression, and thoroughly illustrate its advantages over other baselines.

---

### Decision · Program_Chairs · 2024-01-16

Accept (spotlight)